# Monocyte-secreted Wnt reduces the efficiency of central nervous system remyelination

Bianca M. Hill[1,2,3☯], Rebecca K. Holloway[1,2,3☯], Lindsey H. Forbes[4,5☯], Claire L. Davies[5☯], Jonathan K. Monteiro[1,2,3], Christina M. Brown[4,5], Jamie Rose[4,5], Neva Fudge[6], Pamela J. Plant[1], Ayisha Mahmood[4,5], Koroboshka Brand-Arzamendi[1,2], Sarah A. Kent[4,5], Irene Molina-Gonzalez[4,5], Stefka Gyoneva[7], Richard M. Ransohoff[7,8], Brian Wipke[7,9], Josef Priller[4,10,11,12], Raphael Schneider[1,2], Craig S. Moore[6], Veronique E. Miron [1,2,3,4,5]*

1 Keenan Research Centre for Biomedical Science of St. Michael's Hospital, Toronto, Ontario, Canada, 2 BARLO Multiple Sclerosis Centre, St. Michael's Hospital, Toronto, Ontario, Canada, 3 Department of Immunology, University of Toronto, Toronto, Ontario, Canada, 4 United Kingdom Dementia Research Institute at The University of Edinburgh, Edinburgh, Scotland, United Kingdom, 5 Centre for Discovery Brain Sciences, Chancellor's Building, The University of Edinburgh, Edinburgh, Scotland, United Kingdom, 6 Division of BioMedical Sciences, Faculty of Medicine, Memorial University of Newfoundland, St. John's, Newfoundland, Canada, 7 Previously at Biogen Ltd, Cambridge, Massachusetts, United States of America, 8 Third Rock Ventures, Boston, Massachusetts, United States of America, 9 Manifold Bio, Boston, Massachusetts, United States of America, 10 Centre for Clinical Brain Sciences, Chancellor's Building, The University of Edinburgh, Edinburgh, Scotland, United Kingdom, 11 Department of Psychiatry and Psychotherapy, Klinikum rechts der Isar, School of Medicine, Technical University of Munich, Munich, Germany, 12 Neuropsychiatry and Laboratory of Molecular Psychiatry, Charité-Universitätsmedizin Berlin and DZNE, Berlin, Germany

☯ These authors contributed equally to this work.
* veronique.miron@unityhealth.to

## Abstract

The regeneration of myelin in the central nervous system (CNS) reinstates nerve health and function, yet its decreased efficiency with aging and progression of neurodegenerative disease contributes to axonal loss and/or degeneration. Although CNS myeloid cells have been implicated in regulating the efficiency of remyelination, the distinct contribution of blood monocytes versus that of resident microglia is unclear. Here, we reveal that monocytes have non-redundant functions compared to microglia in regulating remyelination. Using a transgenic mouse in which classical monocytes have reduced egress from bone marrow ($Ccr2^{-/-}$), we demonstrate that monocytes drive the timely onset of oligodendrocyte differentiation and myelin protein expression, yet impede myelin production. Ribonucleic acid sequencing revealed a Wnt signature in wild-type mouse lesion monocytes, which was confirmed in monocytes from multiple sclerosis white matter lesions and blood. Genetic or pharmacological inhibition of Wnt release by monocytes increased remyelination. Our findings reveal monocytes to be critical regulators of remyelination and identify monocytic Wnt signaling as a promising therapeutic target to inhibit for increased efficiency of CNS regeneration.

**Data availability statement:** RNA sequencing data is available on GEO with the accession number GSE121514. Raw flow cytometry data is available on Flow Repository with the ID number FR-FCM-Z9ZJ. Source data underlying the figures are available in S1 Data.

**Funding:** The study was funded by the John David Eaton Chair in Multiple Sclerosis Research and Hall-Sloan Multiple Sclerosis Basic Science Research fund (VEM, St.Michael's Hospital Foundation; https://stmichaelsfoundation.com/), the Multiple Sclerosis Society of the United Kingdom (VEM, JP, 126, https://www.mssociety.org.uk/), a Medical Research Council Senior Non-Clinical Fellowship (VEM; MR/V031260/1, https://www.ukri.org/councils/mrc/), a Medical Research Council Career Development Award (VEM, MR/M020827/1, https://www.ukri.org/councils/mrc/), and Biogen (VEM, https://www.biogen.com/). Employees of Biogen participated in study design and editing of the manuscript but did not participate in data collection and analysis, or decision to publish. All other funders did not participate in study design, data collection and analysis, decision to publish, or preparation of the manuscript.

**Competing interests:** The authors have declared that no competing interests exist.

**Abbreviations:** ANOVA, analysis of variance; CNS, central nervous system; dpl, days post-lesion; FACS, fluorescence-activated cell sorting; FC, fold change; GFP, Green Fluorescent Protein; IPA, Ingenuity Pathway Analysis; LPC, lysophosphatidyl choline; MAG, myelin-associated glycoprotein; MBP, myelin basic protein; MS, multiple sclerosis; OPC, oligodendrocyte progenitor cell; PFA, para-formaldehyde; RNA, ribonucleic acid; VOOM, variance modeling at the observational level; WT, wild type.

## Introduction

Following damage to myelin ("demyelination") in the central nervous system (CNS), the reinstatement of myelin around neuronal axons ("remyelination") restores trophic/metabolic support and rapid nerve conduction. However, with aging and progression of neurodegenerative diseases such as multiple sclerosis (MS), the efficiency of remyelination is decreased, leading to axonal dysfunction and/or loss [1,2]. Current regenerative therapies being trialed to treat MS are focused on enhancing the generation of myelin-forming oligodendrocytes [3]; however, the lack of therapeutic strategies to regulate the efficiency and extent of the subsequent remyelination itself highlights the pressing need to better understand the cellular and molecular mechanisms underpinning this process. Following experimentally-induced demyelination, CNS myeloid cells have beneficial roles early in remyelination through secretion of factors that support the differentiation of oligodendrocyte progenitor cells (OPCs) into new myelin-forming oligodendrocytes, and phagocytosis of myelin debris that otherwise impedes this process [4–8]. However, with aging, these functions are impaired [4,9–11], contributing to inefficient remyelination. CNS lesion myeloid cells encompass resident populations (e.g., microglia) and infiltrating monocytes, which, once the latter differentiate into macrophages, are indistinguishable from one another based on morphology and immunostaining for marker expression [12]. Consequently, microglia and monocyte-derived macrophages are often analyzed as one population, with the respective contribution of each subset to remyelination being poorly understood [13,14]. With the advent of sequencing technologies, recent studies have demonstrated that microglia undergo transcriptional and functional changes during de- and remyelination in mouse models and MS lesions [15–22]. Conversely, the role of monocytes in remyelination is less well-defined [13,14]. Although remyelination in middle-aged mice can be improved by young monocytes provided through parabiosis [9], the mechanistic role of endogenous monocytes in regulating remyelination is unclear. Here, we asked whether transcriptomic analysis and functional manipulation of monocytes could reveal their specific roles in remyelination, thereby addressing the long-standing question as to whether CNS myeloid cell populations have distinct contributions to this process.

## Results

### Monocytes regulate the efficiency of CNS remyelination

To study the specific contribution of monocytes to remyelination, we used a focal lesion model whereby demyelination is induced by stereotaxic injection of the myelin toxin lysophosphatidyl choline (LPC) into the brain white matter tract, the corpus callosum, of young adult mice [4,18,23]. This model is optimal to investigate remyelination specifically, as the well-characterized location and timing of regenerative responses occur without concomitant myelin damage or complicating autoimmunity. We assessed the density of monocytes in lesions at key time points when remyelination is initiated, at 10 days post-lesion (dpl), and complete, at 21 dpl (Fig 1A). This was achieved using a reporter mouse in which selective expression of the chemokine receptor CCR2 by the subtype of monocytes attracted to injured tissue ("classical") drives RFP expression ($Ccr2^{RFP/+}$) (Fig 1B), whereas microglia are RFP negative [24,25]. We detected RFP+ cells in lesions at both time points, yet numbers significantly decreased as remyelination proceeded (Fig 1C and 1D). RFP+ cells were not detected in unlesioned contralateral corpus callosum, indicating recruitment is specific to injured tissue (Fig 1D). We confirmed the presence of monocytes at their peak abundance (10 dpl) via flow cytometry of lesions by selecting for $CD11b^+$ myeloid cells, excluding $Ly6G^+$ neutrophils and $CD3^+$ T cells, then using

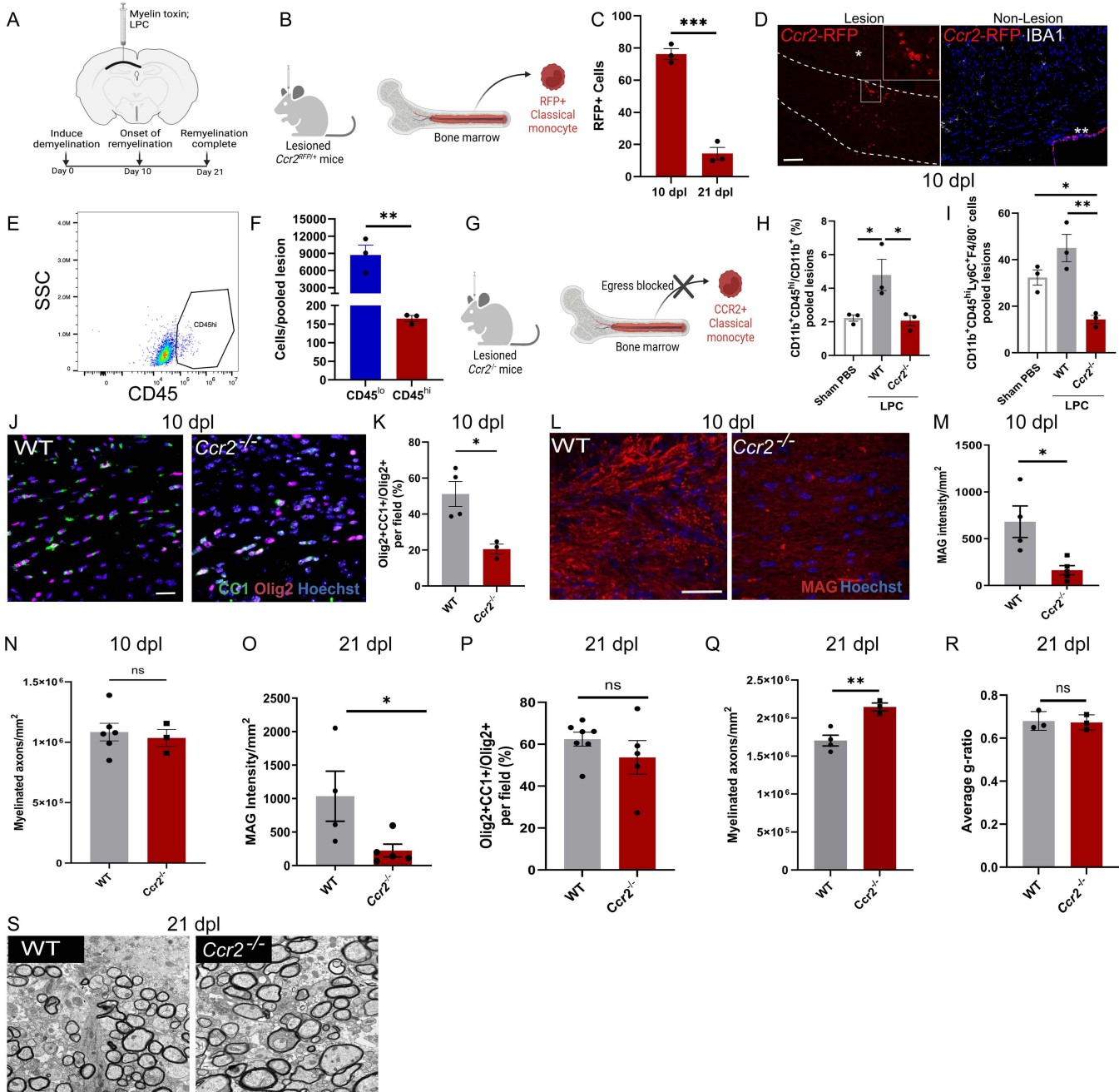

**Fig 1. Classical monocytes regulate central nervous system remyelination.** (A) Schematic of in vivo LPC-induced focal demyelinated lesions in corpus callosum of adult mice. (B) Schematic of use of lesioned *Ccr2*$^{RFP/+}$ reporter mice to track classical monocyte entry into brain lesions. (C) Mean numbers of *Ccr2*-RFP+ cells per lesion image at 10 and 21 days post-lesion (dpl) ± S.E.M. ***$P$ = 0.0003; unpaired 2-tailed Student $t$ test. $n$ = 3 mice/group. (D) Representative immunofluorescent images of *Ccr2*-RFP+ cells in a corpus callosum lesion (left, outlined), *indicates needle tract. Non-lesion corpus callosum (right) stained with IBA1 (white) and counterstained with Hoechst (blue), **indicates choroid plexus monocyte-derived macrophages. Scale bar, 100 μm. (E) Flow cytometry plot demonstrating differential CD45 expression in CD11b$^+$ Ly6G$^-$ CD3$^-$ cells, used to distinguish between lesion microglia (CD45$^{lo}$) and monocytes (CD45$^{hi}$) at 10 dpl. SSC: side scatter. Two mice pooled. (F) Mean numbers of lesion CD45$^{lo}$ microglia and CD45$^{hi}$ monocytes ± S.E.M. at 10 dpl; **$P$ = 0.0080; unpaired 2-tailed Student $t$ test. $n$ = 6 mice/group, 2 mice pooled per replicate. (G) Schematic of use of *Ccr2*$^{-/-}$ mice to impair the egress of classical monocytes from the bone marrow into the blood and subsequently into brain lesions. (H) Mean proportion of monocytes (CD11b+ cells which are CD45$^{hi}$) ± S.E.M. in sham PBS-injected corpus callosum, and lesions from wild type (WT) and *Ccr2*$^{-/-}$ mice. $n$ = 6 mice/group, 2 mice pooled per replicate. One-way analysis of variance (ANOVA) with Tukey's multiple comparisons; *$P$ = 0.0445 (sham vs. WT LPC), *$P$ = 0.0356 (WT vs. *Ccr2*$^{-/-}$ LPC), I) Mean number of monocytes (CD11b$^+$ CD45$^{hi}$ Ly6C$^{hi}$ F4/80$^-$ cells) ± S.E.M. in sham PBS-injected corpus callosum, and lesions from WT and *Ccr2*$^{-/-}$ mice. $n$ = 6 mice/group, 2 mice pooled per replicate. One-way ANOVA with Tukey's multiple comparisons; **$P$ = 0.0040 (WT LPC vs. *Ccr2*$^{-/-}$ LPC), *$P$ = 0.0439 (sham

PBS vs. *Ccr2*$^{-/-}$ LPC). (J) Representative immunofluorescent labeling of oligodendrocyte lineage cells (Olig2+, red) which are mature (CC1+, green) in WT and *Ccr2*$^{-/-}$ mice at 10 dpl, counterstained with Hoechst. Scale bar, 20 μm. (K) Mean percentage of Olig2$^+$ cells which are CC1$^+$ per field ± S.E.M. in WT and *Ccr2*$^{-/-}$ mice at 10 dpl; *P = 0.0156; unpaired 2-tailed Student *t* test. *n* = 3–4 mice/group. (L) Representative immunofluorescent labeling of myelin-associated glycoprotein (MAG, red) in WT and *Ccr2*$^{-/-}$ mice at 10 dpl, counterstained with Hoechst. Scale bar, 20 μm. (M) Quantification of MAG intensity/mm$^2$ within lesioned corpus callosum at 10 dpl ± S.E.M. in WT and *Ccr2*$^{-/-}$ mice, *P = 0.0135; unpaired 2-tailed Student *t* test. *n* = 4–5 mice/group. (N) Mean number of myelinated axons/mm$^2$ at 10 dpl ± S.E.M. in WT and *Ccr2*$^{-/-}$ lesions quantified from electron micrographs. P = 0.6847; unpaired 2-tailed Student *t* test. *n* = 3–6 mice/group. (O) Quantification of MAG intensity/mm$^2$ within lesioned corpus callosum at 21 dpl ± S.E.M. in WT and *Ccr2*$^{-/-}$ mice, *P = 0.0317; Mann–Whitney test. *n* = 4–5 mice/group. (P) Mean percentage of Olig2+ cells which are CC1+ per field ± S.E.M. in WT and *Ccr2*$^{-/-}$ lesions at 21 dpl; P = 0.2879; unpaired 2-tailed Student *t* test. *n* = 5–7 mice/group. (Q) Mean number of myelinated axons in lesions at 21 dpl ± S.E.M. in WT and *Ccr2*$^{-/-}$ lesions; **P = 0.0054; unpaired 2-tailed Student *t* test. *n* = 3–4 mice/group. (R) Average *g*-ratio at 21 dpl ± S.E.M. in WT and *Ccr2*$^{-/-}$ lesions; P = 0.8466; unpaired 2-tailed Student *t* test. *n* = 3 mice/group. (S) Representative electron micrographs of lesioned corpus callosum at 21 dpl in WT and *Ccr2*$^{-/-}$ mice. Scale bar, 2 μm. Mice were 8–12 weeks of age in each group. Source data may be found in S1 Data.

differential expression levels of CD45 to distinguish from microglia – where it has been established previously in models of injured white matter and bone marrow chimeras that monocytes are CD45$^{hi}$ and microglia are CD45$^{lo}$ on a logarithmic scale (Fig 1E; gating strategy in S1A Fig) [24,26–30]. Monocytes were present in lesions, albeit at lower proportions relative to microglia (Fig 1F), but in higher proportions versus sham PBS-injected corpus callosum (Figs 1H and S1A). We confirmed the accuracy of our gating strategy in three ways. First, we demonstrated that the gates distinguish the two cell types clearly in a model where monocyte influx into white matter is robust, following lipopolysaccharide (LPS) injection into the striatum [31] (S1B Fig). Second, we assessed *Ccr2*$^{RFP/+}$ mice for co-expression of RFP and CD45 [24,25] in LPC-induced lesions and found all CD45$^{hi}$ cells to be RFP$^+$ (S1C Fig). Third, we detected expression of a classical monocyte marker, Ly6C, by CD11b$^+$CD45$^{hi}$ lesion cells (S1D and S1E Fig). We next queried the maturation of the cells in lesions. Surprisingly, assessment of macrophage markers by flow cytometry of wild-type mice (F4/80 expression in CD11b$^+$CD45$^{hi}$ Ly6C$^+$ cells) (S1F and S1G Fig), or immunofluorescence analysis of *Ccr2*$^{RFP/+}$ mice (IBA1 expression in RFP$^+$ cells) (S1H and S1I Fig) indicated that approximately 90% of the cells do not express typical macrophage maturation markers in remyelinating lesions. Considering the absence of expression of the monocyte marker Ly6C in a subset of the cells (S1D and S1E Fig), these findings could suggest that monocytes may be in the process of differentiating into macrophages or that the population represents a mixture of undifferentiated and differentiating cells.

Having demonstrated that monocytes are present during remyelination, we then investigated how they influence this process. We assessed the requirement for classical monocytes in remyelination by lesioning transgenic mice lacking expression of CCR2 (*Ccr2*$^{-/-}$), in which their egress from bone marrow is impaired [25,32] (Fig 1G). Microglia do not express CCR2 under homeostasis nor following myelin injury [25], and CCR2-expressing cells do not contribute to CNS-resident macrophages in the corpus callosum [33]. We confirmed that *Ccr2*$^{-/-}$ lesions had reduced recruitment of classical monocytes compared to wild-type lesions, with significantly reduced CD11b$^+$CD45$^{hi}$ cells (Fig 1H) and, of those, reduced Ly6C$^+$ F4/80$^-$ cells (Fig 1I). We observed no difference in lesioning between groups using the myelin protein, myelin basic protein (MBP) (S2A Fig). Next, we investigated how reduced monocyte presence in lesions impacts the regenerative processes regulated by myeloid cells during remyelination: myelin debris clearance and OPC differentiation. Myelin debris had been mostly cleared in both *Ccr2*$^{-/-}$ and wild-type mice at the peak of demyelination, 3 dpl, as indicated by MBP and Oil Red O staining (S2A and S2B Fig). However, differentiation of OPCs into mature oligodendrocytes was significantly impaired in *Ccr2*$^{-/-}$ lesions at the onset of remyelination (10 dpl), with a reduced proportion of oligodendrocyte lineage cells (Olig2+) which were mature (CC1+) (Fig 1J and 1K). We next asked how this affected remyelination; whereas remyelination was initiated normally at 10 dpl in wild-type mice as indicated by expression of the early remyelination marker and myelin

protein myelin-associated glycoprotein (MAG), MAG expression was significantly reduced in lesions from $Ccr2^{-/-}$ mice at this time (Fig 1L and 1M). To complement this, we assessed myelin by electron microscopy-based ultrastructural analysis, first confirming that unlesioned adult $Ccr2^{-/-}$ mice have unaffected myelinated axon density and myelin thickness ("$g$-ratio"; S2C-S2E Fig). Remyelinating lesions surprisingly had no significant difference in the density of myelinated axons between genotypes at 10 dpl (Fig 1N), despite the decrease in mature oligodendrocytes and myelin protein expression in $Ccr2^{-/-}$ lesions at this time; we confirmed lesions were formed in these mice via toluidine blue-stained semi-thin sections (S2F Fig). These findings indicate that when classical monocytes are reduced in remyelinating lesions, oligodendrocyte differentiation is impaired yet more myelin is generated, suggesting that oligodendrocytes may compensate by producing more myelin. Overall, this points to monocytes restricting myelin production during remyelination. To further test this postulate, we investigated lesions when myelin production is normally complete in wild-type controls, at 21 dpl. At this time, MAG remained downregulated in $Ccr2^{-/-}$ lesions versus wild-type control (Fig 1O). However, the proportion of lesion oligodendrocyte lineage cells which had matured in $Ccr2^{-/-}$ mice was now equivalent to that in wild-type controls (Fig 1P). By electron microscopy, we found that $Ccr2^{-/-}$ mice had ~25% increase in number of myelinated axons compared to wild-type controls at 21 dpl (Fig 1Q and 1S). We assessed the structural integrity of this remyelination and found no significant differences from wild-type control in myelin thickness ($g$-ratio; Figs 1R and S2G), nor in frequency of structural abnormalities (S2H and S1I Figs). These findings reveal that, although classical monocytes are required for the timely onset of oligodendrocyte differentiation and myelin protein expression during remyelination, they limit myelin production. The latter suggests that monocytes reduce the efficiency of remyelination itself.

## Monocytes have a Wnt signature during remyelination

Having identified that monocytes regulate remyelination efficiency, we next investigated the molecular mechanisms influencing this function by transcriptomic analysis of lesion monocytes in comparison with microglia. Given the low number of lesion monocytes, we achieved this by performing ultra-low input bulk ribonucleic acid (RNA) sequencing of these cells subsequent to fluorescence-activated cell sorting (FACS)-sorting of CD11b$^+$CD3$^-$ Ly6G$^-$ CD45$^{hi}$ cells at 10 dpl (S1A Fig). We confirmed enrichment of microglia-associated genes in the CD45$^{lo}$ gate relative to the CD45$^{hi}$ gate (e.g., $P2ry12$, $Tgfbr1$, $Sall1$, $Tmem119$) by RNA sequencing (Fig 2A) and quantitative polymerase chain reaction (qPCR) (S3A Fig); our findings are in line with some of these genes having previously been shown to be expressed in monocyte-derived macrophages albeit at a relatively lower level than in microglia (e.g., $P2ry12$, $Tmem119$), whereas $Sall1$ is never expressed by monocytes [34–37]. Furthermore, we found enrichment of monocyte-enriched genes in the CD45$^{hi}$ cells (S3B Fig). Our gating strategy successfully excluded other neural and immune cell populations (S3C Fig). A low level of transcript for the myelin/oligodendrocyte gene $Mbp$ was detected in microglia but not monocytes (S3C Fig), consistent with previous observations of myelin transcripts normally located in oligodendrocyte processes being found in microglia during CNS injury, likely reflecting phagocytosis [38]. However, no significant difference in expression of genes associated with phagocytosis was detected between microglia and monocytes (S3D Fig). Lesion monocytes and microglia showed significantly different transcriptomes (S1 Sheet). Analysis of the top 10% of all expressed genes revealed that monocytes and microglia share expression of 1,566 genes (Fig 2B), including myeloid cell identity genes such as $Csf1r$, $Itgam$, $Cx3cr1$, and $Ptprc$, as expected. However, of these top expressed genes, monocytes, and microglia both express 934 genes that are not highly expressed by the other population (Fig 2B; top 50 indicated in Fig 2C). We

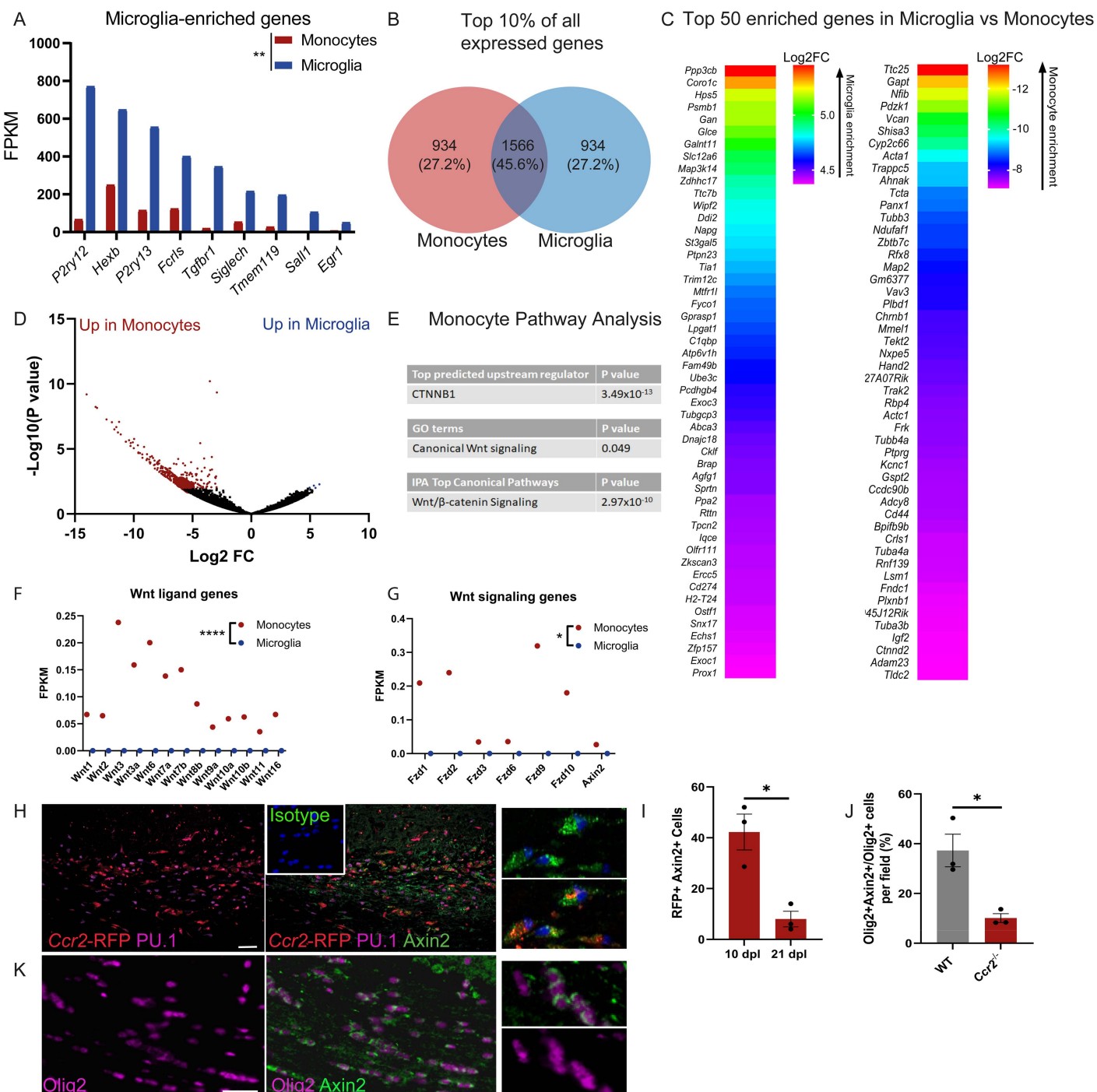

**Fig 2. Monocytes have a Wnt signature during remyelination.** (A) Mean expression in Fragments per Kilobase of Transcript per Million mapped reads (FPKM) of microglia-enriched genes in lesion monocytes (red) and microglia (blue) at 10 dpl. **$P = 0.0026$; 2-tailed paired Student $t$ test. $n = 2$–3 mice/group. (B) Venn diagram of top 10% of all expressed genes in lesion microglia and monocytes at 10 dpl. (C) Log2 fold change (FC) in top 50 enriched genes in microglia (left) and monocytes (right). Values were generated by comparing microglia transcriptomes to monocyte transcriptomes; the positive values indicate an upregulation (enrichment) in microglia, whereas negative values indicate a downregulation in microglia, therefore, an enrichment in monocytes. Red indicates highest enrichment and purple indicates the lowest enrichment. $n = 2$–3 mice/group. (D) Volcano plot showing significance as –Log10 transformed $P$ values against Log2FC. Black indicates no significant change, red indicates genes upregulated in monocytes vs. microglia, and blue indicates genes upregulated in microglia vs. monocytes. $P$ value cutoff 0.01. $n = 2$–3 mice per group. (E) Pathway analysis of genes upregulated in monocytes vs. microglia at 10 dpl. (F) Average FPKM of Wnt ligand genes in monocytes (red) and microglia (blue) in lesions at 10 dpl. ****$P < 0.0001$; one sample $t$ test against microglia value of 0. (G) Average FPKM of Wnt signaling genes in monocytes (red) and microglia (blue) in lesions at 10 dpl. *$P = 0.0152$; one sample $t$ test against microglia value of 0. (H) RFP+ lesion monocytes (red) expressing

Axin2 (green) at 10 dpl, counterstained with myeloid cell nuclear marker PU.1 (purple). Inset, Axin2 isotype control. Scale bar, 100 μm. (I) Mean number of RFP+ cells expressing Axin2 ± S.E.M. at 10 and 21 dpl in $Ccr2^{RFP/+}$ reporter mouse lesions. *$P = 0.00111$; unpaired 2-tailed Student $t$ test. $n = 3$ mice/time point. (J) Mean percentage of Olig2+ cells expressing Axin2 ± S.E.M. at 21 dpl in WT and $Ccr2^{-/-}$ mice; *$P = 0.0162$; unpaired 2-tailed Student $t$ test. $n = 3$ mice/group. (K) Oligodendrocyte lineage cells (Olig2+; purple) expressing Axin2 (green) at 21 dpl. Scale bar, 50 μm. Mice were 8–12 weeks of age in each group. Source data may be found in S1 Data.

interrogated the differentially expressed genes (DEGs) between the two cell types using a stringent significance threshold ($P < 0.01$) and observed that whereas only 4 genes were upregulated in microglia versus monocytes, monocytes upregulated 457 genes relative to microglia (Fig 2D). This indicated that these myeloid cell populations have differing transcriptomes during remyelination, with monocytes primarily upregulating genes relative to microglia.

To further interrogate the molecular mechanisms regulating monocyte function during remyelination, we performed pathway analysis on DEGs. The Wnt signaling regulator β-catenin (CTNNB1) was predicted to be a top upstream regulator of monocyte DEGs, and gene ontology (GO) term and Ingenuity Pathway Analysis (IPA) suggested significant regulation of the Wnt signaling pathway (Fig 2E). This is of interest given the previous association of Wnt signaling with monocyte responses to injury in other organs [16,39–41] and with the regulation of remyelination [42–44]. In monocytes, we detected expression of 13 of the 19 Wnt ligands (Fig 2F), 6 of the Wnt receptors (*Fzd1, 2, 3, 6, 9, 10*), and a major downstream Wnt target and signaling component (*Axin2*) (Fig 2G), whereas these gene products were not detectable in microglia. These findings suggest active autocrine Wnt signaling within monocytes during remyelination. To validate this postulate, we assessed Axin2 protein expression in lesion monocytes using $Ccr2^{RFP/+}$ reporter mice. Axin2+ RFP+ cells were detected at both 10 and 21 dpl, significantly decreasing over time as remyelination proceeds (Fig 2H and 2I). We also detected Wnt-responsive oligodendrocyte lineage cells in remyelinating lesions (Axin2+ Olig2+), which were significantly decreased in $Ccr2^{-/-}$ lesions (Fig 2J and 2K). Altogether, these findings suggest that monocytes have a Wnt signature during remyelination and that Wnt responsiveness is both autocrine and paracrine within lesions.

## Wnt signature in monocytes in multiple sclerosis

We queried the translational relevance of our findings by investigating changes in monocyte Wnt activity in MS white matter lesions. As it is challenging to confidently distinguish between monocytes and microglia by immunostaining of human post-mortem tissue, we took advantage of a recently published single-nuclei RNA sequencing dataset of the rim of mixed MS lesions, which are sites of transition from efficient to poor remyelination [16,45]. Data-mining the genes significantly enriched in the 2 classical monocyte clusters (renamed clusters 1 and 2) relative to other immune cell clusters revealed that ~50% of genes are known to either be regulated by Wnts or regulate Wnt signaling (Fig 3A and 3B). We then asked whether such changes could also be detected in MS blood monocytes in association with clinical progression, as poor remyelination is observed in individuals with high disability [46], by assessing a highly-expressed subset of these genes using qPCR (Figs 3C and S3E). Of the 18 genes showing a ≥2-fold increase in expression in progressive MS monocytes over the average in healthy controls, 14 genes were significantly increased in cases with high clinical disability (secondary progressive multiple sclerosis [MS]; EDSS ≥ 6) versus those with low clinical disability (relapse-remitting MS; EDSS ≤ 2) (Fig 3C). This was not a result of higher age in the progressive MS group (S1 Table). Our results indicate that a Wnt signature is already partially induced in MS monocytes prior to CNS entry, which is increased in association with clinical disability progression.

## A  MS lesion monocyte cluster 1

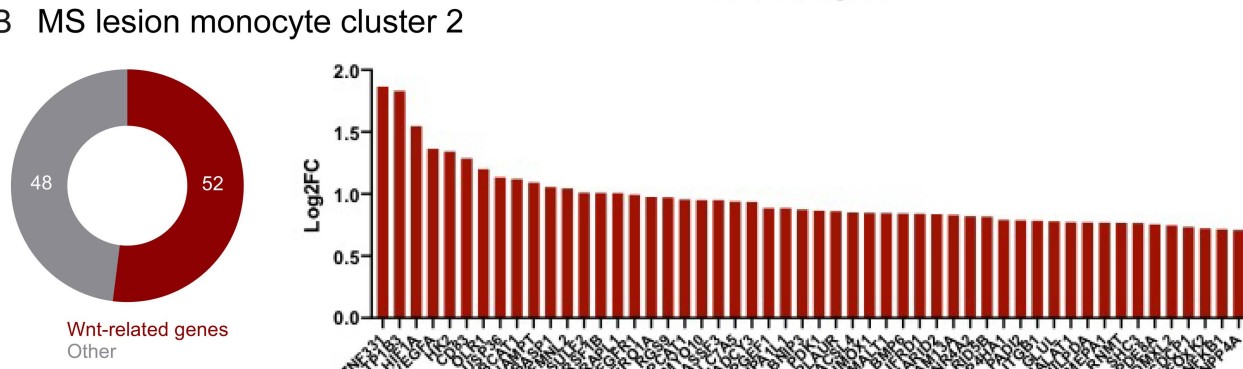

## B  MS lesion monocyte cluster 2

## C  MS blood monocytes

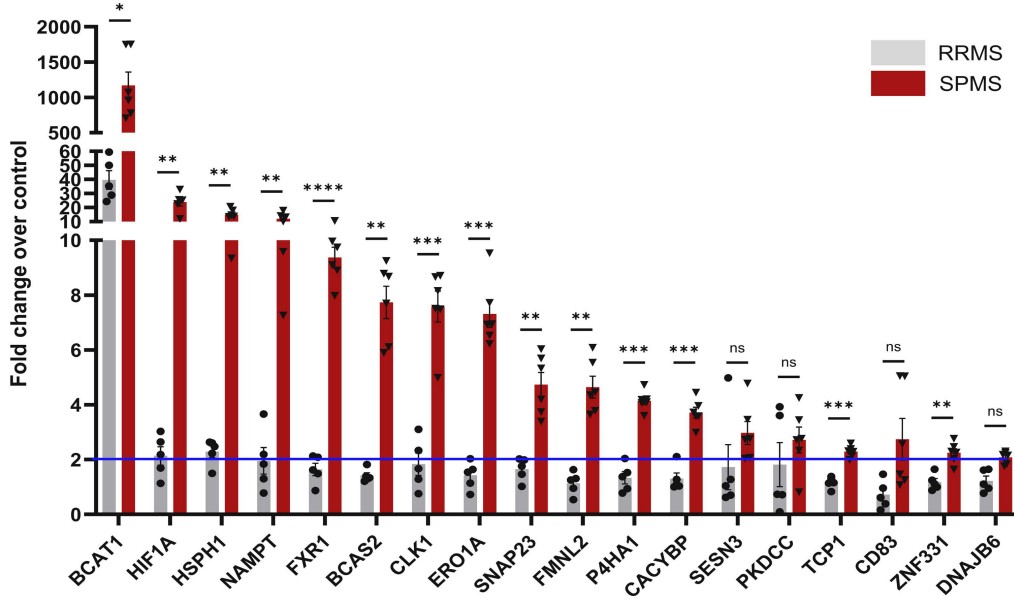

**Fig 3. Wnt signature in MS monocytes.** (A) Expression of Wnt-related genes in monocyte cluster 1, sequenced in the rim of mixed active MS lesions, represented as percentage of total cluster genes (left) and Log2 fold change (FC) (right). (B) Expression of Wnt-related genes in monocyte cluster 2, sequenced in the rim of mixed active MS lesions, represented as percentage of total cluster genes (left) and Log2FC (right). (C) Mean expression of Wnt-related genes in blood monocytes from individuals with relapse-remitting MS (RRMS; EDSS ≤2; $n = 5$) and secondary progressive MS (SPMS; EDSS ≥6; $n = 6$), which showed a ≥2-fold increase over healthy control ($n = 4$), represented as fold change over control. Brown-Forsythe

and Welch ANOVA with Dunnet's T3 multiple comparisons test; *P* = 0.0202 (*BCAT1*), 0.006 (*HIF1A*), 0.0058 (*HSPH1*), 0.0097 (*NAMPT*), <0.0001 (*FXR1*), 0.0014(*BCAS2*), 0.003 (*CLK1*), 0.0002 (*ERO1A*), 0.0050 (*SNAP23*), 0.0012 (*FMNL2*), 0.0002 (*P4HA1*), 0.0002 (*CACYBP*), 0.0001 (*TCP1*), 0.0084 (*ZNF331*). Source data may be found in S1 Data.

## Monocyte Wnt signaling reduces the efficiency of remyelination

We next explored the role of Wnts in monocyte function during remyelination by inhibiting their Wnt activity. First, we used a mouse model previously employed to assess the requirement of Wnt signaling in monocytes in the repair of other organs [41], where Wnt activity is conditionally knocked out by eliminating expression of the Porcupine enzyme (*Porc*) involved in the post-translational modifications of Wnts required for their release. In the absence of an existing transgenic mouse in which Cre recombinase is specifically active in monocytes without potential confounding activity in other immune cells [47,48], we used the *Csf1r*-iCre line which is effective in knocking out *Porc* in myeloid cells [41] (Fig 4A). Although this approach targets both microglia and monocytes, we did not detect a Wnt signature in microglia during remyelination. We found that in floxed controls, remyelination occurred at a slower pace than expected, with minimal remyelination at 14 dpl likely resulting from the Friend leukemia virus B background (versus the standard C57Bl/6) [4,18]. Despite this finding, at this time, *Csf1r*-iCre;*Porc*^fl/fl^ mice showed enhanced remyelination compared to floxed controls, as indicated by increased expression of the late-stage remyelination marker myelin basic protein (MBP; Fig 4B and 4C). However, there was no significant difference in the proportion of mature oligodendrocytes (CC1+ Olig2+) in *Csf1r*-iCre;*Porc*^fl/fl^ versus floxed control lesions at 14 or 21 dpl, indicating increased remyelination independent of an impact on oligodendrocyte differentiation (Fig 4D and 4E).

Second, we aimed to specifically target monocytic Wnt activity. We asked whether we could influence the efficiency of remyelination by intravenously supplementing classical monocytes – with and without *ex vivo* Wnt inhibition – to lesioned *Ccr2*^−/−^ mice, in which endogenous circulating monocytes are significantly reduced (Fig 4F). To track injected monocytes, we isolated cells from the bone marrow of adult mice in which myeloid cells express Green Fluorescent Protein (GFP) (*Csf1r*-eGFP), via magnetic-activated cell sorting (S3F Fig). To inhibit the Wnt pathway, we treated isolated cells *ex vivo* with a small molecule inhibitor of Porcupine, C59, prior to extensive washing and tail vein injection of 500,000 cells in 100 μL of PBS. GFP+ cells were observed in lesions of mice injected with either vehicle-treated or C59-treated monocytes, with no significant difference in densities between groups (Figs 4G and S3G). GFP+ cells were IBA1 negative, indicating maintenance of monocyte identity without macrophage differentiation (S3H Fig), consistent with our findings in wild type and *Ccr2*^RFP/+^ mice (S1H and S1I Fig). C59 impaired Wnt signaling as demonstrated by reduced Axin2 staining (32 ± 3% Axin2+GFP+/ total GFP+ cells versus 60 ± 3% in vehicle control, *P* = 0.0023; Fig 4G). Injection of C59-treated monocytes into *Ccr2*^−/−^ mice significantly increased remyelination at 10 dpl compared to vehicle-treated monocyte injection, evidenced by increased MAG signal (Fig 4H and 4I). However, there was no significant difference in oligodendrocyte differentiation between treatment groups (Fig 4J and 4K). Overall, these findings reveal that monocyte-derived Wnt decreases the efficiency of remyelination, independently of impacting oligodendrocyte differentiation.

## Discussion

Here, we uncover specific roles for monocytes in regulating the efficiency of remyelination. We discovered that monocytes support oligodendrocyte differentiation and myelin protein expression, yet impede myelin production in a Wnt-dependent manner (Fig 4L).

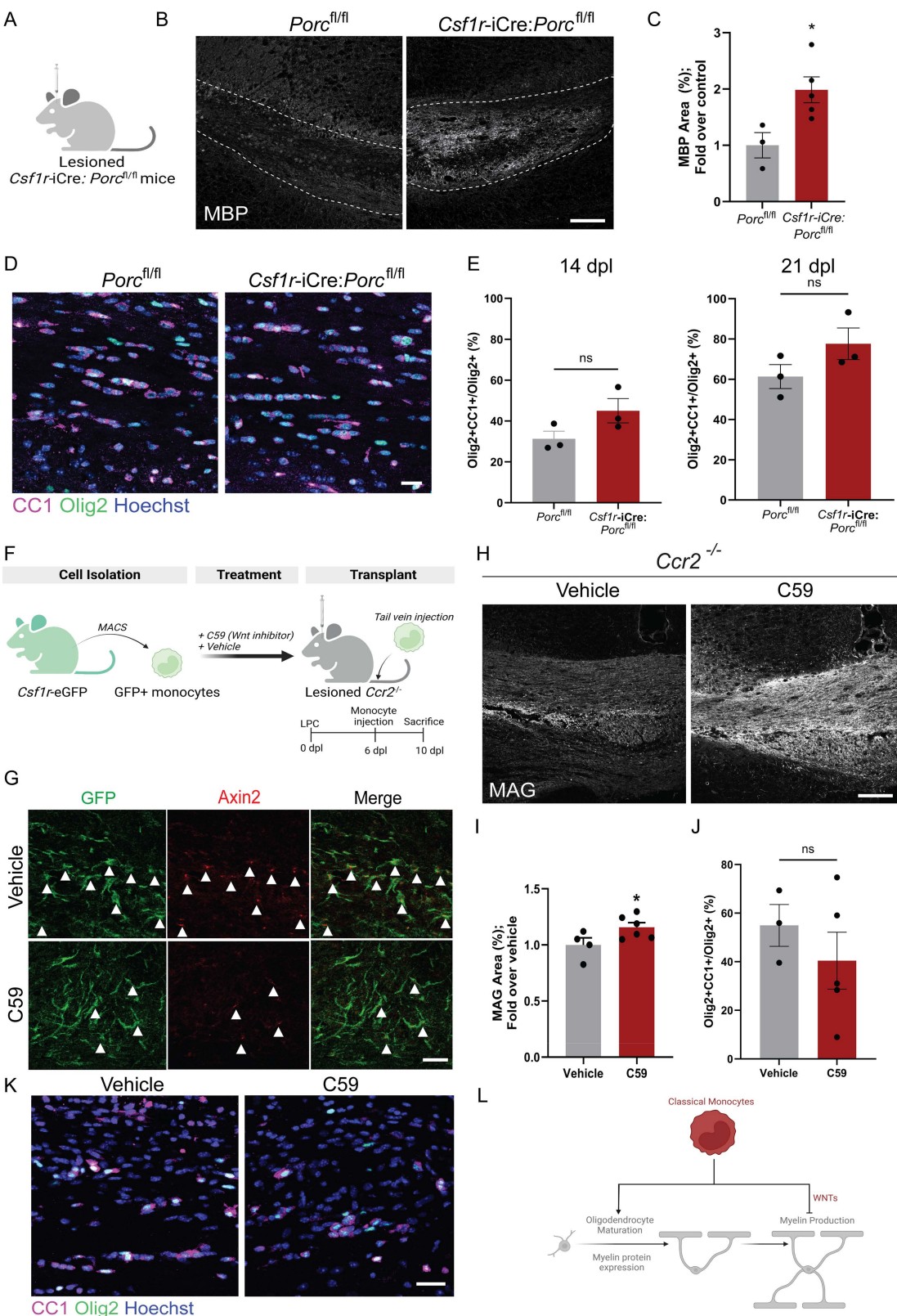

**Fig 4. Wnt signaling in monocytes regulates remyelination efficiency.** (A) *Csf1r*-iCre;*Porc*^fl/fl were lesioned to assess the impact on remyelination. (B) Representative images of myelin basic protein (MBP) expression within lesions (outlined) at 14 dpl in *Csf1r*-iCre;*Porc*^fl/fl compared to *Porc*^fl/fl controls. Scale bar, 25 μm. (C) Mean percentage area of MBP staining ± S.E.M. in

corpus callosum of *Csf1r*-iCre;*Porc*fl/fl compared to *Porc*fl/fl controls at 14 dpl. *P = 0.0130, one-tailed *t* test against control value of 1. *n* = 3–5 mice/group. (D) Representative immunofluorescent labeling of oligodendrocyte lineage cells (Olig2+, green) which are mature (CC1+, magenta) in *Csf1r*-iCre;*Porc*fl/fl compared to *Porc*fl/fl control mice at 14 dpl, counterstained with Hoechst. Scale bar, 20 μm. (E) Mean percentage of Olig2+ cells which are CC1+ per field ± S.E.M. in *Csf1r*-iCre;*Porc*fl/fl compared to *Porc*fl/fl controls at 14 and 21 dpl; *P = 0.1210 (14 dpl), 0.1741 (21 dpl); unpaired 2-tailed Student *t* test. *n* = 3 mice/group. (F) Schematic of transplant of C59- or Vehicle-treated GFP+ monocytes into lesioned *Ccr2*−/− mouse circulation. (G) GFP+ transplanted cells detected in lesions of *Ccr2*−/− mice, following *ex vivo* treatment with vehicle or C59, and stained for Axin2 (red), at 10 dpl. Double positive cells indicated with arrows. Scale bar, 25 μm. (H) Representative images of MAG expression within lesions at 10 dpl in lesioned *Ccr2*−/− mice transplanted with C59- or vehicle-treated monocytes. Scale bar, 25 μm. (I) Mean percentage area of MAG staining ± S.E.M. in C59-treated monocyte transplanted *Ccr2*−/− mice at 10 dpl, normalized to vehicle-treated monocyte transplanted control. *P = 0.0131, one sample *t* test against value of 1. *n* = 4–6 mice/group. (J) Mean percentage of Olig2+ cells which are CC1+ per field ± S.E.M. in C59- or vehicle-treated monocyte transplanted *Ccr2*−/− mice at 10 dpl. *P = 0.4217; unpaired 2-tailed Student *t* test. *n* = 3–5 mice/group. (K) Representative immunofluorescent labeling of oligodendrocyte lineage cells (Olig2+, green) which are mature (CC1+, magenta) in Vehicle-treated and C59-treated monocyte transplanted mice at 10 dpl, counterstained with Hoechst. Scale bar, 20 μm. (L) Graphical abstract of results: Classical monocytes support oligodendrocyte differentiation and myelin protein expression, yet impede myelin production during remyelnation in a Wnt-dependent manner. *Csf1r*-iCre; *Porc*fl/fl were 8–14 weeks old. *Ccr2*−/− in transplant experiments were 8–16 weeks old. Source data may be found in S1 Data.

Our finding that monocytes support oligodendrocyte differentiation at the initiation of remyelination is consistent with prior literature suggesting their secretion of growth factors influencing oligodendrogenesis early-on in remyelination, such as IGF-1 and TGF-β [9,49]. However, our discovery that monocytes impede myelin production, especially in the final stage of remyelination, is novel. Interestingly, our observed increased myelin production concomitant with a downregulation of MAG protein when monocytes are constitutively reduced (in lesioned *Ccr2*−/− mice) is consistent with myelination proceeding in excess in MAG global knockout mice (*Mag*−/−) [50,51]. Conversely, our finding contrasts previous work where depletion of circulating monocytes using clodronate liposomes resulted in reduced late-stage remyelination (21 dpl) [49,52]; in light of our data, we postulate that this result may have been consequent to the reported rebound increase in lesion myeloid cells subsequent to the short-term depletion [49], potentially increasing the number of monocytes present to impair remyelination. We acknowledge other potential differences with our study, including sex of the animals and increased monocyte recruitment to spinal cord versus brain lesions.

By sequencing monocytes in remyelinating lesions and performing functional interventions, we uncovered that they impede myelin production during remyelination via Wnt signaling. Whereas monocytic Wnt signaling promotes the repair of the liver [39], gut [41], kidney [53], and vascular network [40], our results are in line with Wnt signaling being known to impair remyelination [42–44]. The downstream activation of the Wnt pathway in monocytes during remyelination suggests autocrine signaling. In other tissues, Wnt signaling in monocytes regulates gene expression to influence phagocytosis and release of both pro- and anti-inflammatory cytokines [54]. We did not observe changes in myelin debris clearance in *Ccr2*−/− lesions following demyelination, demonstrating that the Wnt-driven impact of monocytes on remyelination is independent of this process and pointing to the possible release of a Wnt-regulated factor restricting myelin production. In parallel, monocyte-released Wnts may directly influence other neural cells during remyelination. We observed that microglia do not express Wnt receptors nor downstream effectors during remyelination thereby ruling out their response to monocyte-derived Wnts, in contrast to the microglial response to Wnts following developmental white matter injury [55]. However, oligodendrocyte lineage cells showed Axin2 reactivity during remyelination, which was in part dependent on monocytes. Wnt signaling in OPCs is known to impede remyelination by reducing oligodendrocyte differentiation [42–44], although we found no significant reduction in mature oligodendrocytes when monocyte-mediated Wnt release was blocked. Rather, we saw an increase in remyelination,

which indicates a new role for Wnt signaling in regulating myelin production by oligodendrocytes during remyelination. Consistent with our mouse functional studies, we detected a Wnt signature in monocytes in MS brain lesions transitioning to poor remyelination efficiency [16,45], and in blood monocytes in progressive MS when remyelination often fails [46]. Although future work is required to understand which of the numerous Wnt gene targets are functionally implicated in impeding remyelination, the genes identified as being significantly altered in MS monocytes may in future serve as blood-based proxies for remyelination efficiency, pending validation. Altogether, this suggests that Wnt signaling in monocytes could contribute to impaired remyelination in MS and may represent a promising biomarker for poor remyelination efficiency.

We identified that monocytes have non-redundant roles from microglia during remyelination, as we observed altered oligodendroglial and myelin responses in the absence of monocytes which were not compensated for by the presence of microglia. Although this result is perhaps surprising given the low number of monocytes in brain lesions, very low numbers of infiltrating regulatory T lymphocytes and pericytes have been shown to regulate remyelination efficiency, via supporting oligodendrogenesis [56,57]. Our finding complements the known differential roles of distinct myeloid cell populations in the induction of autoimmune-mediated demyelination in the experimental autoimmune encephalomyelitis model, with microglia being protective whereas monocytes initiate damage [29,38,58,59]. Microglia and monocytes communicate with each other following CNS damage, with young monocytes supporting acquisition of a regenerative state by old microglia and myelin debris phagocytosis in parabiotic experiments [4,9], and microglia limiting the endogenous recruitment of monocytes into demyelinated lesions [12]. Given our findings, the latter may serve to limit monocyte entry to increase remyelination efficiency, however, future work is needed to fully elucidate how each myeloid cell population regulates the regenerative capacity of the other. Nonetheless, our finding of monocyte-specific regulation of remyelination, independent of the roles of microglia, has important implications for the development of regenerative therapeutics. Single-cell RNA sequencing of MS acute lesion biopsies suggested that monocyte contributions are variable yet could account for up to 60% of lesion myeloid cells [15], thus representing a significant population for therapeutic targeting. The opportunity to influence monocytes circulating in the blood is attractive, where taking advantage of the homing of classical monocytes to injured tissue eliminates the need for development of therapies that cross the blood-brain barrier. Indeed, a proof-of-concept study showed that transplanted genetically-modified blood immune cells, an approach which preferentially targeted monocytes, could be used to deliver a pro-regenerative factor to demyelinated lesions to enhance remyelination [60].

In summary, our study highlights the classical monocyte as a critical cell type that reduces remyelination efficiency. We propose that inhibiting monocytic Wnt signaling may represent a novel therapeutic strategy to enhance remyelination efficiency with progression of MS.

## Materials and methods

### *In vivo* focal CNS white matter demyelination

Experiments were performed according to ARRIVE2 guidelines under a project license issued to VEM (PP8451532) under the Animals (Scientific Procedures) Act 1986 and reviewed then approved by the Animal Welfare Ethical Review Body of the University of Edinburgh and the United Kingdom Home Office, or were approved by the Animal Care Committee at St. Michael's Hospital (protocol numbers 230, 231, 232) in accord with the Canadian Council on Animal Care (CCAC). Mice were provided food and water *ad libitum* and exposure to 12-h light/dark cycles. Demyelinated lesions were induced in the corpus callosum of adult male

(8–17 weeks) C57BL/6J mice (Jackson Laboratory 000664), *Ccr2*$^{-/-}$ mice (B6.129S4-*Ccr2tm1lfc*/J, Jackson Laboratory 004999), *Csf1r-iCre;Porc*$^{fl/fl}$ and *Porc*$^{fl/fl}$ mice [41] (Jackson Laboratory 021024), and *Ccr2*$^{RFP/+}$ mice (B6.129[Cg]-*Ccr2*tm2.1lfc/J, Jackson Laboratory 017586), by stereotaxic injection of 2 μL of 1% lysolecithin (LPC; vol/vol, Sigma-Aldrich L4129) using a Hamilton syringe at coordinates from Bregma: −1.2 mm caudal, −0.5 mm lateral, −1.4 mm ventral, as previously described [4,14,18,23]. Mice were sacrificed at various days post-lesion (dpl). To confirm our gating strategy to distinguish between microglia and monocytes, LPS was injected stereotaxically into the striatum (coordinates from Bregma: 0.0 mm caudal, −2.0 mm lateral, 2.5 mm ventral), and mice were sacrificed at 3 days post-injection. Mice used for FACS or flow cytometry were intracardially perfused with ice-cold 1X phosphate-buffered saline (PBS; Sigma-Aldrich, D8537). Mice used for immunofluorescence were intracardially perfused with PBS, then 4% paraformaldehyde (PFA, wt/vol; Sigma-Aldrich, P6148 or BioShop Canada, PAR070); brains were post-fixed overnight and cryoprotected in sucrose (BioShop Canada, SUC700) before OCT embedding (VWR, 361603E or Fisher, 23-730-571) and storage at −20 °C.

## Immunofluorescent staining of mouse tissue

Frozen coronal brain sections (10 μm) were air-dried for 15 min and blocked for 1 hour room temperature with PBS containing 5% horse serum (Gibco, 2,605,008) and 0.3% Triton-X 100 (Fisher Scientific, 10254583). For Axin2, Olig2, and PU.1 staining, heat-activated antigen retrieval was carried out by microwaving sections on a medium setting in 10 mM citrate buffer (pH 6.0; Vector laboratories, H-3300) and heating for 30 min at 60 °C in an oven. Sections were washed in 1X TBS (Fisher Scientific, BP24711) with 0.001% Triton-X-100 (Fisher Scientific, 10254583) prior to incubation with primary antibodies. For myelin and RFP stains, sections were incubated with ice-cold methanol at −20 °C for 10 min and washed in PBS prior to primary antibody incubation. Primary antibodies were applied overnight at 4 °C in a humid chamber and included rat-anti-MBP (Millipore, AB5,864, 1:250), rabbit anti-Axin2 (1:50; Abcam, ab18582 or 1:200; Abcam, ab32197), mouse anti-MAG (1:100; Sigma-Aldrich, MAB1567), rabbit anti-RFP (1:100; Abcam, ab62341 or ab185921), rabbit anti-Olig2 (1:100; Sigma-Aldrich, AB9610), goat anti-IBA1 (1:200; Abcam, ab5076), rat-anti-PU.1 (1:100; Biolegend, 681202), and mouse anti-APC (CC1; 1:500, Abcam, ab16794). For Axin2, Olig2, and RFP, primary antibodies were amplified using tyramides. Briefly, after washing in 1XTBS with 0.001% Triton-X to remove the primary antibody, sections were incubated with the species-specific HRP-conjugated secondary antibody (Vector Laboratories, anti-rabbit VECTMP740115 or anti-mouse VECTMP740215) before being washed again in 1XTBS with 0.001% Triton-X and incubated for 10 min in a humid chamber with either Opal 520 (1:100; Akoya Biosciences, FP1487001KT), Opal 570 (1:100; Akoya Biosciences, FP1488001KT), or Opal 650 (1:100; Akoya Biosciences, FP1496001KT). After a final washing step, the sections were co-stained with additional primary antibodies listed above, after which sections were incubated with Alexa fluor-conjugated secondary antibodies (1:500; Life Technologies) for 2 h at 20–25 °C in a humid chamber. Sections were then counterstained with Hoechst (Sigma-Aldrich, B2261 or Thermo Fisher Scientific, H1339), washed in PBS, and cover slipped using Fluoromount-G (Invitrogen, 00-4958-02). Tissue was imaged using an Olympus 3i Spinning Disk confocal microscope, a Leica SP8 confocal microscope, a Zeiss LSM 900, or a Zeiss wide field microscope.

## Tissue processing and electron microscopy

Mice were intracardially perfused with 4% PFA (wt/vol) and 2% glutaraldehyde (vol/vol, Sigma-Aldrich G5882) in 0.1 M phosphate buffer. Tissue was then post-fixed overnight in 1% glutaraldehyde in 0.1 M phosphate buffer at 4 °C. About, 1 mm tissue sections were processed

into araldite resin blocks. Semi-thin (1 μm) sections were cut using a microtome and stained using a solution of 1% toluidine blue and 2% sodium borate prior to imaging on a Zeiss Axio microscope using brightfield to identify areas containing lesioned corpus callosum. Ultra-thin sections (60 nm) were subsequently cut of the lesioned corpus callosum, and stained with uranyl acetate and lead citrate before imaging on a JEOL Transmission Electron microscope. Axon size and $g$-ratios were measured using ImageJ (FIJI) software with a minimum of 100 axons measured per animal, and number of myelinated axons was counted using ImageJ (FIJI) software with a minimum of 10 images per animal analyzed.

## Myeloid cell detection and isolation from focal CNS remyelinating lesions

Monocytes and microglia were isolated from focal remyelinating lesions of corpus callosum as previously described [18,61]. Briefly, following perfusion with ice-cold PBS, brains were extracted and stored on ice in Hanks' Balanced Salt Solution (HBSS) without $Ca^{2+}$ and $Mg^{2+}$ (Gibco, 14,175,095) with 10% FBS (Gibco, 10,500,064). Lesioned corpus callosum was dissected out from a 2 mm coronal section then homogenized manually, with a minimum of 40 passes using a 2 mL Dounce (DWK Life Sciences, 10,530,762). Following filtration (75 μm filter; Pierce, 87,791) cells were spun at 600 g for 5 min with brake. Cells were resuspended in 100% fetal bovine serum (FBS; Gibco, 10,500,064) and 33% isotonic Percoll (1:10; SLS, 17,089,102 or Cytiva, 17-0891-01). 1 mL of 10% FBS was carefully overlaid to create a concentration gradient, and the cells were spun for 15 min at 800 g at 4 °C without brake. The cell pellet was washed in FACS buffer (1% FBS, 1 μm EDTA in PBS) and spun for 10 min at 600 g at 4 °C, then resuspended in FACS buffer. Cells isolated from lesions were added to a 96-well plate and centrifuged at 400 g for 3 min. Cells were incubated in anti-mouse CD16/32 Fc-block (1:200; BD, 553142 or 1:400; BioLegend, 101319) and viability dye (1:1,000; eBioscience, 65086613) diluted in FACS buffer and incubated on ice for 20 min. Fluorescently-conjugated antibodies were added directly to the samples without washing and incubated on ice for 30 min. Antibodies included anti-mouse CD11b-PeCy7 (1:100; eBioscience, 25-0112-81 or 1:1,000, BioLegend, 101216, Clone M1/70), anti-mouse CD45-BV605 (1:300; Biolegend, 103139, Clone 30-F11), Ly6G-PerCP Cy5 (1:200, BioLegend, 127615, Clone 1A8) or Ly6G-PE/Dazzle 594 (1:600, BioLegend, 127648, Clone 1A8), Ly6C-FITC (1:600; BD, 553104, Clone AL-21), CD3-APC (1:200, BioLegend, 100235, Clone 17A2 or A700; 1:1,000, BioLegend, 100216, Clone 17A2) and F4/80-APC (1:600; BioLegend, 123116, Clone BM8). Cells were centrifuged at 400 g for 3 min resuspended in FACS buffer and passed through a 30 μm filter (Sysmex, 04-004-2326) prior to sorting (BD FACSAria Fusion) or analysis by flow cytometry (BD LSR Fortessa or CytoFLEX-LX). Cells were sorted into 1.5 mL Eppendorf tubes coated with 100% FBS then kept on ice. FlowJo was used for post-acquisition data analysis. To facilitate visualization of monocyte numbers in lesions, 3 mice were pooled for $Ccr2^{RFP/+}$ lesion analysis, and 2 mice were pooled repeated 3 times for a total of 6 mice/group for wild-type sham PBS-injected, wild-type LPC-injected, and $Ccr2^{-/-}$ lesion analyses.

## RNA extraction and ultra-low input RNA sequencing

The average cell yield was 5,777 microglia and 38 monocytes per mouse. We sequenced an average of 4 ng of RNA for microglia and 1.6 ng of RNA from monocytes, using ultra-low input RNA sequencing which is optimized for samples with as little as 0.5 ng; this was sufficient for our purposes as we detected high expression of myeloid cell-associated genes in both microglia and monocyte populations and differential expression of genes known to be enriched in microglia versus monocytes. Isolated cells were centrifuged at 800 g for 5 min, resuspended in 350 μL RLT Plus buffer with β-mercaptoethanol (QIAGEN, 1053393), and

centrifuged at 8,600 g for 2 min using QIAshredder tubes (QIAGEN, 79656). RNA was extracted using the AllPrep DNA/RNA/miRNA kit (QIAGEN, 80224) as per the manufacturer's instructions. Quantity and quality of RNA were determined using the Bioanalyzer 2100 (Agilent) and RNA 6000 Pico kit (Agilent, 5067−1513) as per the manufacturer's instructions. cDNA production and library preparation were performed by BGI (Hong Kong). Briefly, double-stranded cDNA was prepared according to Ovation RNA-Seq System V2 (Tecan/NuGen, 7102−32), as per the manufacturer's instructions. The amplified cDNA was combined with End Repair Mix (QIAGEN, Y9140-LC-L) and incubated at 20 °C for 30 min. End-repaired DNA was purified with AxyPrep Mag polymerase chain reaction (PCR) clean-up kit (Axygen, MAG-PCR-CL-50), then combined with A-Tailing Mix (Enzymatic) at 37 °C for 30 min. Adaptors (Invitrogen) were ligated to the Adenylate 3′ ends DNA, incubated at 16 °C for 16 h, and purified with AxyPrep Mag PCR clean-up kit. The adapter-ligated DNA fragments were selected based on the insert size and enriched by PCR amplification; the PCR products (final library) were purified with AxyPrep Mag PCR clean-up kit. The final library was quantitated in two ways: The average molecule length was determined using the Bioanalyzer 2100 using the DNA 1000 kit (Agilent, 5067−1504), and the library was quantified by real-time quantitative PCR (qPCR) (TaqMan Probe). The qualified libraries were amplified using cBot (Illumina) to generate the cluster on the Flow Cell (HiSeq 4000, Illumina). An average of 40M clean 100 paired-end reads were obtained per sample. Data was processed to remove adaptors, and low-quality reads were removed from raw reads.

## Bioinformatics

Raw data analysis was conducted by Fios Genomics (Edinburgh, UK). Briefly, data were pre-processed and aligned to the mouse genome (GRCm38) using STAR aligner, and number of mapped read pairs per gene was quantified based on GENCODE vM12 annotation. Quality control of samples was conducted using the array Quality Metrics package in Bioconductor. Data were normalized using trimmed mean of M values (TMM) and transformed using variance modeling at the observational level (VOOM) to Log2-counts per million (Log2- CPM) or Fragments per kilobase of transcript per million mapped reads (FPKM). Linear modeling was used for group comparisons using Limma and incorporating precision weights from VOOM to identify DEGs ($P < 0.01$). Functional analysis was conducted using GO term analysis, Venny2.1, and Ingenuity Pathway Analysis (IPA; QIAGEN).

## Real-time quantitative polymerase chain reaction (RT-qPCR) on mouse cells

Real-time (RT) qPCR was run using BioRad Custom PrimePCR plates, as per the manufacturer's instructions. Briefly, cDNA was synthesized using 5x iScript Advanced Reaction Mix and iScript Advanced Reverse Transcriptase (BioRad, 1,708,890) at 46 °C for 20 min then 95 °C for 1 min. cDNA samples were mixed with 2x iTaq Universal SYBR-green Supermix (BioRad, 1725120) and iScript Reverse Transcriptase (BioRad, 1708840). The RT-qPCR was performed at 50 °C for 10 min, 95 °C for 1 min, 95 °C for 15 sec (40 cycles), and 60 °C for 60 sec (40 cycles). Data was analyzed using CFX manager and is represented as $2^{-\Delta Ct}$.

## Bone-marrow-derived monocyte isolation and transplantation

Monocytes were isolated from the bone marrow of 8−12 week-old MacGreen (*Csf1r-eGFP*) male mice using magnetic cell sorting (MACS) according to the manufacturer's instructions (Miltenyi Biotec), and cells were kept on ice or at 4 °C throughout. Briefly, bone marrow cells were collected from the femur and tibia bones of MacGreen mice by flushing the shaft with a

syringe and needle containing buffer solution (0.5% bovine serum albumin; Miltenyi Biotec, 130-091-376, and 2 mM EDTA; Thermo Fisher, AM9260G, in PBS). Cells were blocked and stained using a monocyte biotin-antibody cocktail (Miltenyi Biotec) for 5 min at 4 °C. Cells were subsequently washed with buffer solution and centrifuged for 10 min at 300 g. The cell pellet was resuspended in buffer containing anti-biotin microbeads (Miltenyi Biotec, 130-091-256) and incubated for 10 min at 4 °C before proceeding to magnetic separation. Separation was carried out by applying the cell suspension to an LS column (Miltenyi Biotec, 130-042-401) attached to a MidiMACS magnetic separator (Miltenyi Biotec, 130-042-302) and collection of the flow-through containing the enriched monocyte fraction. Monocytes were then treated with 10 nM C59 (Tocris, 5,148) or DMSO vehicle, diluted in PBS for 25 min at 4 °C. Cells were centrifuged for 10 min at 300 g and resuspended in PBS prior to injection into the tail vein of $Ccr2^{-/-}$ mice. Each mouse received 1 injection containing 500,000 cells in 100 µL of PBS. Identity of isolated cells following MACS was confirmed using flow cytometry. Cells were incubated in anti-mouse CD16/32 Fc-block (Clone 93, 1:200; BD, 55,314) diluted in FACS buffer, and incubated on ice for 10 min. Fluorescently-conjugated antibodies were added directly to the samples and incubated on ice for 30 min. Antibodies included anti-rat CD11b-BV650 (1:100; BioLegend, 101239), anti-mouse CD115-BV605 (1:100; BioLegend, 135517), rat-anti Ly6G-BV510 (1:200; Biolegend, 127633), anti-rat Ly6C-PB (1:100; Biolegend, 128014), and rat-anti CD45-PE/Dazzle 594 (1:100; Biolegend, 103139). Cells were centrifuged at 400 g for 5 min and resuspended in FACS buffer then passed through a 30 µm filter prior to sorting (BD FACS Aria Fusion).

## Data-mining published MS monocyte transcriptomes

Datasets were obtained from a published study of single-nuclei RNA sequencing of MS chronic active (i.e., "mixed") lesions [16]. In their study, the authors subclustered immune cells (*CSF1R+*, *C3+*, *CD74+*, *RUNX1+*) and lymphocytes (*PTPRC+*, *CD2+*, *SKAP1+*), then further subclustered these cells. Two clusters were annotated as monocytes based on upregulation of *CD83* and lower expression (negative Z scores) of microglia-enriched genes *SALL1* and *TMEM119* compared to all immune cells, including microglia. Peripheral immune cell (PBMC) signatures were further mapped onto a multimodal peripheral blood mononuclear cell reference dataset to confirm identities (https://satijalab.org/seurat/v4.0/reference_mapping.html); the majority of the monocytes expressed *CD14* rather than *CD16*, therefore representing classical monocytes. We mined the transcriptomes of both clusters for genes that are known to be regulated by Wnt or regulate Wnt signaling.

## Human monocyte isolation, RNA extraction, and qPCR

Participants with relapse-remitting or secondary progressive MS (using the 2017 revised McDonald criteria) were recruited from the MS clinics at NL Health (St John's, NL, Canada) as part of the Health Research Innovation Team in Multiple Sclerosis study. Participants provided written consent with prior approval by the provincial Health Research Ethics Board. Age and sex-matched healthy control subjects were also recruited. Blood was drawn via venipuncture from four healthy controls, five patients with relapsing-remitting multiple sclerosis and six patients with secondary progressive MS; all participants were not on treatment, were female, and aged between 26 and 66 years old (S1 Table). Immediately after blood collection, PBMCs were isolated using SepMate PBMC isolation tubes (STEMCELL Technologies, 85460) and a Ficoll-Paque (Cytiva, 17-1440-03) density gradient. PBMCs were washed twice with PBS, counted, and cryopreserved in freezing media (20% DMSO, 10% DMEM, 70% FCS) and liquid nitrogen. CD14+ monocytes were positively selected to 95%–98% purity from

thawed PBMCs using MACS anti-human CD14 microbeads, LS columns, and QuadroMACS separators (Miltenyi Biotec, 130-050-201, 130-042-401). The CD14+ cells were stored at −80 °C in 500 μL QIAzol Lysis Reagent (Qiagen, 79306). RNA was extracted using an RNeasy Micro kit (Qiagen, 74004). Briefly, 100 μL of chloroform (Fisher Scientific, AC423,555,000) was added to the sample, agitated for 2–3 min and centrifuged at 4 °C at 12,000 g for 15 min. The aqueous layer was removed and transferred to an RNase-free 1.5 mL centrifuge tube. The RNA was precipitated with 350 μL 70% ethanol and added to a spin column. The samples were centrifuged, washed, treated with DNase, and eluted with 14 μL RNase-free water as per the manufacturer's instructions. A total of 200 ng RNA and 250 ng of random primers were used to synthesize complementary DNA (cDNA) using the M-MLV reverse transcriptase kit (Invitrogen, 28025013). The cDNA was stored at −80 °C. Gene-specific pre-amplification of the resulting cDNA was performed by mixing 5 μL of cDNA with custom primers (330,241, QIAGEN) before being placed in a real-time cycler (Biorad C1000 Thermal Cycler). Side reaction reducer was used to eliminate residual primers from the pre-amplification step before samples were stored at −20 °C. The PCR components mix for each sample was then generated using 102 μL of the amplified cDNA plus RT2 SYBR-green ROX qPCR Mastermix (330521 QIAGEN) before being loaded into a 384-well custom RT2 PCR array (330181, QIA-GEN). Plates were run on QuantStudio 7 (Applied Biosystems) using SYBR as the method of detection. Following qPCR, genes were excluded from analysis if the difference in CT values between technical duplicates was greater than 1. Data is represented as fold difference in relative quantity over healthy control, calculated as $2^{\Delta Ct \text{ (target gene in control – MS)}}$/ Geometric mean of $2^{\Delta Ct \text{ (housekeeping gene in control – MS)}}$; the housekeeping genes selected were those stably expressed across groups (*ACTB* and *HSP90AB1*).

## Statistics

Statistical analysis was performed using GraphPad Prism software. Gaussian distribution was analyzed using the Shapiro–Wilks test or Kolmogorov–Smirnov test. Data are presented as mean ± standard error of the mean (S.E.M.). Data were analyzed using unpaired Student *t* test, Wilcoxon paired *t* test, Mann–Whitney test, One-way analysis of variance (ANOVA) with Tukey's post hoc test, or Two-way ANOVA with Sidak's multiple comparisons test. Human monocyte qPCR data were analyzed using a Brown Forsyth Welch ANOVA with Dunnet's T3 multiple comparisons post-test. Values of $P \leq 0.05$ were considered statistically significant.

## Supporting information

**S1 Fig. Detection of lesion monocytes.** (A) Gating strategy for flow and fluorescence-activated cell sorting of myeloid cells in lysophosphatidylcholine (LPC)-induced lesions vs. sham PBS-injected corpus callosum, gating on live cells, CD11b+ cells, excluding Ly6G+ granulocytes and CD3+ T cells, and distinguishing microglia from monocytes based on loga-rithmic differential expression of CD45. Plots represent pooling of 2 mice per condition. SSC: side scatter. (B) Validation of gating strategy in a model in which monocytes (CD11b+CD3− Ly6G− CD45hi) are abundant, following lipopolysaccharide injection into the striatum. (C) Validation of gating strategy using LPC-injected *Ccr2*RFP/+ reporter mice in which all CD45hi cells are RFP+, representing pooling of 3 mouse lesions. (D) Flow cytometry plot indicat-ing Ly6C expression by CD11b+CD45hi cells, representing a pooling of 2 mouse lesions. (E) Mean percentage of CD45hi cells which express Ly6C ± S.E.M. Each data point represents a pooling of 2 mice, with *n* = 6 mice total. (F) Flow cytometry plot indicating that the major-ity of CD11b+CD45hi Ly6C+ cells in lesions are negative for the macrophage marker F4/80, representing a pooling of 2 mouse lesions. (G) Mean number of lesion CD11b+CD45hi Ly6C+

cells which are F4/80 positive vs. negative ± S.E.M. **$P = 0.0011$, 2-tailed Student $t$ test. Each data point represents a pooling of 2 mice, with $n = 6$ mice total. (H) Mean number of RFP+ cells which are IBA1− (red) or IBA1+ (grey) per field of lesion ± S.E.M. at 10 and 21 dpl. ****$P < 0.0001$ (10 dpl), 0.0177 (21 dpl), 2-way ANOVA with Sidak's multiple comparisons tests. $n = 3$ mice/time point. (I) Representative fluorescent images of $Ccr2^{RFP/+}$ lesions stained for the macrophage marker IBA1 (white). Scale bar, 25 μm. Mice were 8–12 weeks old. Source data may be found in S1 Data.
(TIF)

**S2 Fig. Myelin properties in Ccr2 knockout mice.** (A) Representative images of lesions in wild type (WT) and $Ccr2^{−/−}$ mice as indicated by absence of MBP (red) in the corpus callosum (outlined in white) at 3 dpl. Scale bar, 25 μm. (B) Myelin debris at 3 dpl in WT and $Ccr2^{−/−}$ mice indicated by Oil Red O, counterstained with hematoxylin. Scale bar, 10 μm. (C) Mean number of myelinated axons/mm$^2$ in unlesioned corpus callosum of WT and $Ccr2^{−/−}$ mice. $P = 0.4599$; unpaired Student $t$ test. $n = 3$ mice/group. (D) Average $g$-ratio in unlesioned corpus callosum of WT and $Ccr2^{−/−}$ mice. $P > 0.9999$; unpaired Student $t$ test. $n = 3$ mice/group. (E) Dot plot of $g$-ratio vs. axon diameter (μm) for all axons in unlesioned corpus callosum of WT (grey) and $Ccr2^{−/−}$ mice (red). $n = 100$ axons/mouse/genotype, $n = 3$ mice/group. (F) Representative semi-thin sections of lesions in WT and $Ccr2^{−/−}$ mice at 10 dpl stained for toluidine blue. Scale bar, 100 μm. (G) Dot plot of $g$-ratio vs. axon diameter (μm) for all axons in lesioned corpus callosum of WT (grey) and $Ccr2^{−/−}$ mice (red). $n = 100$ axons/mouse/genotype, $n = 3$ mice/group. (H) Representative electron micrographs of myelin abnormalities in lesions as highlighted in red, including double myelin (i), myelin inclusions (ii), and myelin outfoldings (iii). (I) Average number of abnormal myelin structures/mm$^2$ in WT (grey) and $Ccr2^{−/−}$ (red) lesioned corpus callosum ± S.E.M. $P = 0.5534$; 2-tailed Student $t$ test, $n = 3$ mice/group. Mice were 8–12 weeks old. Source data may be found in S1 Data.
(TIF)

**S3 Fig. RNA sequencing and analysis of lesion monocytes.** (A) Quantitative PCR of microglia signature genes ± S.E.M. in 10 dpl lesion monocytes (red) and microglia (blue). ND = not detected. *$P = 0.0351$; 2-tailed paired $t$ test. $n = 2$–3 mice/group. (B) Log2 fold change (FC) enrichment of monocyte-enriched genes in the CD11b$^+$ CD3$^-$ Ly6G$^-$ CD45$^{hi}$ vs. CD45$^{lo}$ cells. (C) Average FPKM of genes associated with oligodendrocytes (*Mbp*, *Olig2*), neurons (*Rbfox3*), astrocytes (*Aldh1l1*, *Gfap*), vascular cells (*Tie1*, *Cdh5*), granulocytes (*Ly6g*), and T cells (*Cd3*, *Cd4*) in monocytes (red) and microglia (blue) in lesions at 10 dpl. ND = not detected. (D) Average FPKM of phagocytosis-associated genes in monocytes (red) and microglia (blue) in lesions at 10 dpl. $P = 0.0835$; 2-tailed paired $t$ test. (E) Mean expression of Wnt-associated genes in blood monocytes from individuals with relapse-remitting MS (RRMS; EDSS ≤ 2; $n = 5$) and secondary progressive MS (SPMS; EDSS ≥ 6; $n = 6$) which showed a <2-fold increase over healthy control ($n = 4$), represented as FC over control. Brown-Forsythe and Welch ANOVA with Dunnet's T3 multiple comparisons test; $P = 0.0325$ (ATP1B3), 0.0376 (VEGFA). (F) Proportion of cells, indicated as cell frequency (%), from MACS-sorted bone marrow cells prior to transplant into $Ccr2^{−/−}$ mice. Neutrophils (CD45$^+$ CD115$^-$ Ly6G$^{hi}$); MDP; monocyte dendritic cell progenitor (CD45$^+$ CD115$^+$ CD11B$^-$ Ly6G$^{lo}$ Ly6C$^{lo}$); mono: monocytes (CD45$^+$ CD115$^+$ CD11B$^+$ Ly6G$^{lo}$, Ly6C$^{hi}$ vs. Ly6C$^{lo}$). (G) Mean number of GFP+ cells per lesion field in $Ccr2^{−/−}$ mice injected with C59- or vehicle-treated monocytes. $P = 0.4236$; unpaired 2-tailed Student t test. $n = 4$–6 mice/group. (H) Transplanted monocytes (GFP+; asterisk) in lesions positive for Axin2 (red) but negative for IBA1 (white). Mice were 8–12 weeks old. Source data may be found in S1 Data.
(TIF)

**S1 Table. MS blood monocyte sample information.** Information for samples from healthy controls and people with relapse-remitting multiple sclerosis and secondary progressive multiple sclerosis: age, sex (F: female), time since last relapse, and expanded disability status scale (EDSS) score.
(PDF)

**S1 Sheet. RNA sequencing of lesion microglia and monocytes.** Log2 fold change and P values of gene expression comparisons between CD11b$^+$ Ly6G$^-$ CD3$^-$ CD45$^{lo}$ cells (microglia) and CD11b$^+$ Ly6G$^-$ CD3$^-$ CD45$^{hi}$ cells (monocytes) from lysophosphatidyl choline-induced lesions of corpus callosum at 10 days post-lesioning.
(XLSX)

**S1 Data. Source data. Numerical data used to generate the graphs in each of the figure panels, shown in separate tabs.**
(XLSX)

## Acknowledgments

We acknowledge the core facility staff at (i) the Keenan Research Centre for Biomedical Science Core Facilities at St. Michael's Hospital (Monika Lodyga, Caterina Di Ciano-Oliviera, Danielle Bince) for support with flow cytometry, imaging, and surgery, (ii) the SickKids Cellular and Molecular Electron Microscopy facility (Ali Darbandi) for support with electron microscopy, and (iii) the University of Edinburgh (Shonna Johnson, Will Ramsay, Stephen Mitchell) for support with FACS and electron microscopy. We also thank Anne Cotleur, Dr. Barry McColl, Dr. Jonathan Moss, and Fios Genomics for technical assistance, and Dr. Luca Cassetta and Prof. Jeffrey Pollard for scientific discussions. We acknowledge the co-principal investigators (M. Ploughman, C. Moore, F. Clift, and M. Stefanelli) and study participants enrolled in the Health Research Innovation Team in Multiple Sclerosis (HITMS), an MS patient registry/biospecimen repository and partnership between researchers and clinicians at Newfoundland Health and Memorial University of Newfoundland (St. John's, Newfoundland, Canada). All diagrams were created with Biorender.com.

## Author contributions

**Conceptualization:** Bianca M. Hill, Lindsey H. Forbes, Claire L. Davies, Richard M. Ransohoff, Brian Wipke, Josef Priller, Veronique E. Miron.

**Data curation:** Bianca M. Hill, Lindsey H. Forbes, Claire L. Davies, Neva Fudge, Koroboshka Brand-Arzamendi, Veronique E. Miron.

**Formal analysis:** Bianca M. Hill, Rebecca K. Holloway, Lindsey H. Forbes, Claire L. Davies, Pamela J. Plant, Sarah A. Kent, Veronique E. Miron.

**Funding acquisition:** Craig S. Moore, Josef Priller, Veronique E. Miron.

**Investigation:** Bianca M. Hill, Rebecca K. Holloway, Lindsey H. Forbes, Claire L. Davies, Jonathan Monteiro, Christina M. Brown, Jamie Rose, Neva Fudge, Ayisha Mahmood, Koroboshka Brand-Arzamendi, Sarah A. Kent, Irene Molina-Gonzalez, Brian Wipke, Josef Priller, Raphael Schneider, Craig S. Moore, Veronique E. Miron.

**Methodology:** Bianca M. Hill, Rebecca K. Holloway, Lindsey H. Forbes, Claire L. Davies, Koroboshka Brand-Arzamendi, Neva Fudge, Stefka Gyoneva, Richard M. Ransohoff, Brian Wipke, Josef Priller, Raphael Schneider, Craig S. Moore, Veronique E. Miron.

**Project administration:** Richard M. Ransohoff, Brian Wipke, Josef Priller, Raphael Schneider, Craig S. Moore, Veronique E. Miron.

**Supervision:** Raphael Schneider, Craig S. Moore, Veronique E. Miron.

**Writing – original draft:** Veronique E. Miron.

**Writing – review & editing:** Bianca M. Hill, Rebecca K. Holloway, Lindsey H. Forbes, Claire L. Davies, Ayisha Mahmood, Irene Molina-Gonzalez, Stefka Gyoneva, Richard M. Ransohoff, Brian Wipke, Josef Priller, Raphael Schneider, Craig S. Moore, Veronique E. Miron.

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
