## [Editor Report · Decision Letter 0]

22 Jan 2024

Dear Veronica,

Thank you for submitting your manuscript entitled "Monocytes regulate the efficiency of central nervous system remyelination" for consideration as a Short Report by PLOS Biology.

Your manuscript has now been evaluated by the PLOS Biology editorial staff as well as by an academic editor with relevant expertise and I am writing to let you know that we would like to send your submission out for external peer review.

Once your full submission is complete, your paper will undergo a series of checks in preparation for peer review. After your manuscript has passed the checks it will be sent out for review. To provide the metadata for your submission, please Login to Editorial Manager (https://www.editorialmanager.com/pbiology) within two working days, i.e. by Jan 24 2024 11:59PM.

Kind regards,

Luke

Lucas Smith, Ph.D.

Senior Editor

PLOS Biology

lsmith@plos.org

---

## [Decision Letter · Decision Letter 1]

8 Mar 2024

Dear Veronica,

Thank you for your patience while your manuscript "Monocytes regulate the efficiency of central nervous system remyelination" was peer-reviewed at PLOS Biology. Your manuscript has been evaluated by the PLOS Biology editors, an Academic Editor with relevant expertise, and by several independent reviewers.

As you will see in the reviewer reports, which can be found at the end of this email, although the reviewers find the work potentially interesting, they have also raised a substantial number of important concerns. Based on their specific comments and following discussion with the Academic Editor, it is clear that a substantial amount of work would be required to meet the criteria for publication in PLOS Biology. However, given our and the reviewer interest in your study, we would be open to inviting a comprehensive revision of the study that thoroughly addresses the reviewers' comments. Given the extent of revision that would be needed, we cannot make a decision about publication until we have seen the revised manuscript and your response to the reviewers' comments. Your revised manuscript would need to be seen by the reviewers again, but please note that we would not engage them unless their main concerns have been addressed. 

Having discussed the reviews with the Academic Editor, we think you should focus on strengthening and fleshing out the conclusions presented here as the reviewers suggest. For example, we think it will be important to address reviewer 3's concern about the gating strategy - which will possibly require additional experiments. We also note that several reviewers highlight the need to more thoroughly characterize MDM recruitment with and without demyelination and following inhibition of Wnt signaling and we think this would be important. We also agree with Reviewers 2 and 4 that a more thorough analysis of OPCs, OLs and equivalent methodology to assess myelin status at 10 and 21 dpi would be very helpful, and we think that concerns with statistical rigor will need to be carefully addressed. 

As a last note - we would not strictly require the generation of additional data from an EAE model or more in-depth analysis of involvement of specific Wnts in this revision, as suggested by the reviewers. While we think these suggestions are interesting, we think they are likely beyond the scope of the current work, particularly considering that it is submitted as a 'short report'. 

We appreciate that the requests, detailed above, represent a great deal of extra work, and we are willing to relax our standard revision time to allow you 6 months to revise your study. Please email us (plosbiology@plos.org) if you have any questions or concerns, or envision needing a (short) extension.

**IMPORTANT - SUBMITTING YOUR REVISION**

*Resubmission Checklist*

*Published Peer Review*

*PLOS Data Policy*

*Blot and Gel Data Policy*

Sincerely,

Luke

Lucas Smith, Ph.D.

Senior Editor

PLOS Biology

lsmith@plos.org

REVIEWS:

Reviewer #1: In this manuscript by Forbes et al., the authors report a newly discovered role for blood monocyte-derived macrophages (MDMs) in regulating myelin regeneration within the central nervous system (CNS) following local insult/injury. Their study aims to show that this role is unique to these CNS macrophages and not microglia and modulated via the WNT signaling pathway using a combination of genetic and adoptive transfer models. Analysis of public patient genomic data also suggest a role for WNT signaling dysregulation in the pathobiology of multiple sclerosis (MS). Taken together, these findings are particularly exciting for their implications that MDMs represent a candidate for cell-therapy based treatment of demyelinating disorders. The data are clear, concise, the interpretations of the results measured, and overall, the manuscript is very polished. There are a few lingering concerns and constructive comments however, that if addressed can help to improve the manuscript and strengthen their findings:

1) Despite the strong connection between the role of WNT signaling pathway in slowing CNS MDM- regulated remyelination, the authors provide little context or discussion of downstream mechanism or effector function of Wnt signaling in slowing of remyelination, and how their results expand on previous findings. In addition, what Wnts and effector cells (or autologous signaling) do the authors think are in play? This would strengthen the Discussion section, and better contextualize the impact of their findings, and suggest future directions for this work. 

2) To link Wnt signaling more firmly to MDM-mediated inhibition of remyelination, there are two potential approaches the authors can take 

a. The Csf1r-iCre;Porcfl/fl mouse is knocksout WNT signaling in both MDMs or microglia. A prudent control experiment to more robustly show that WNT signaling in microglia has no effect on remyelination is a microglia-specific Porc knockout, perhaps by using the Tmem119-Cre line available through the Jackson Labs. This is very time intensive and may not be required if the authors expand their adoptive transfer approaches

b. An alternative explanation for why Wnt inhibitor pretreatment resulted in less remyelination after MDM engraftment is that there are fewer MDMs in the lesion site. Quantifying the total number of C59-treated Csf1r-eGFP+ MDMs engrafted (via quantifying total eGFP+ cells counterstained with a nuclear marker) in/around regions of high MAG or MBP staining would better demonstrate that WNT signaling inhibition rather than lack of engrafted monocytes drives increased remyelination.

As for a few minor comments/edits:

1) Typo on page 3, line 17: "With advent" should be changed to "With the advent".

2) Add the ages of the mice for all animal-based experiments within the Results section (or within the figure legends for each corresponding dataset) as opposed to this only being listed in the Methods section increases the rigor.

3) In the legend for Figure 1B, describe the abbreviation in the y-axis.

4) The current design/presentation of the heatmaps in Figure 2C is confusing and difficult to interpret. Use of a less subtle gradient scale for the FC values, as well as a description in the figure legend for how to interpret said scales for both the microglial and MDM datasets would improve readability and interpretation.

5) For the parts of a whole graphs in Figures 3A &B, add the percentile values over the "Wnt-related genes" and "Others" portions of the graphs.

6) For the analyses discussed and presented in Figure 3C, is there a reason only the group analysis of all 5 WNT genes in MS patients v. Ctrls is showed, as opposed to presenting the results of the individual comparisons of expression levels for MS v. Ctrl for each gene? If that statistical data is available, please report it in the supplement.

7) List the alpha values used for all statistical analyses in the Methods sections.

8) List catalogue/product numbers for all antibodies used across experiments and the strain numbers for all mouse lines purchased from the Jackson Lab and used in this study within the Methods section.

9) Page 12, Methods, line 20: add the concentration for the PBS used here.

10) Typo on page 13, Methods, line 14: change "Ultrathin" to "Ultra-thin". 

In conclusion, this manuscript by Forbes et al. presents well-conducted and rigorous studies with several exciting implications for WNT signaling-associated proteins as potential biomarkers for demyelinating diseases and avenues for developing novel cell therapeutics for treating such disorders. This manuscript would interest a wide readership from neuroscience and immunology-related fields and those interested in translational science and novel therapeutic avenues for neurodegenerative disease.

Reviewer #2: This manuscript by Forbes, investigated the role of monocytes in regulating the efficiency of CNS remyelination. The authors utilized a genetic approach to block monocyte egressing to understand the role of blood-derived monocytes during remyelination in a toxin-induced demyelination animal model. The results revealed that the monocyte elimination resulted in a functional switch in a tempo-dependent manner during remyelination. Finally, they identified overactivation of Wnt signaling in monocytes contributes to the inhibitory role of monocytes in the demyelination model.

The role of monocytes in remyelination has been previously reported (Glia, 2001 35(3), 204-212; Neurobiology of Disease,2005 18(1), 166-175). This manuscript reported a tempo change of monocyte functions---switching from pro-myelination at the early phase to anti-myelination at the late stage. This phenotype is intriguing, but the authors did not provide substantial evidence to understand this tempo function change. Secondly, the high tone of Wnt signaling in demyelinating lesions has been demonstrated repeatedly, how much do the monocytes/macrophages contribute to the Wnt tone in the lesions. 

Major concerns: 

1. This manuscript reported blocking monocyte egress by using the Ccr2 knockout mice leads to a pro-myelination effect at the late stage. The key message of this manuscript is partially contradictory to previous reports showing that elimination of monocytes/macrophages resulted in hindered remyelination in demyelinating animal models (Glia, 2001 35(3), 204-212; Neurobiology of Disease, 2005 18(1), 166-175). How to understand this discrepancy? Is this related to different approaches to manipulating monocytes?

2. The functional switch of monocytes during remyelination is interesting. However, the authors did not explain this phenotype. For example, is the functional switch related to different degrees of Wnt tones? Different numbers of infiltrated monocytes/macrophages?

3. High-tone Wnt signaling has been demonstrated repeatedly in demyelinating lesions that can inhibit remyelination (Figure 1). How much do monocytes contribute to the level of Wnt tone in the lesions? Given the low density of monocytes/macrophages in the lesions, do the monocytes/macrophages directly regulate OPC differentiation and myelination? Or indirectly through other cell types, like microglial cells?

4. Single-cell sequencing analysis revealed that the Wnt signaling was altered in the MS monocytes of the blood samples of MS, indicating Wnt changes before infiltration into CNS. An EAE model may better mirror the change of Wnt signaling in the monocytes.

5. How do monocytes regulate OPC differentiation and myelination remains puzzling from the current data set, as the changes in OPC differentiation and myelination were not thoroughly examined. For example, the densities of mature oligodendrocytes and OPCs were not assessed in Figure 1 E, G. Similarly, the myelin, OL, and OPC changes were not examined in Figure 4.

Reviewer #3: Several studies have demonstrated the importance of microglia in promoting myelin turnover in demyelination lesions. Nevertheless, the role of monocyte-derived macrophages in these lesions is still poorly understood. This study by Forbes and colleagues addresses this question using CCr2-/- mice, which feature impaired egress of monocytes from the bone marrow. Using the LPC model of focal demyelination, the authors observed delayed remyelination in knockout mice compared to wild-type mice at day 10 post-injection. However, knockout mice successfully remyelinated the lesion by day 21. The authors found that CD11b+CD45hi cells, which they refer to as MDMs, are enriched for WNT signature genes. Consequently, they hypothesized that WNT signaling could play a role in remyelination. The authors blocked WNT signaling using two complementary approaches: 1) Cre-driven deletion of Porc in monocytes/macrophages; 2) adoptive transfer of monocytes treated ex-vivo with a WNT inhibitor into Ccr2-/- lesioned mice. Both approaches resulted in increased remyelination compared to controls. The authors concluded that MDMs infiltrating the demyelination lesion slow the remyelination process through WNT signaling. While the findings presented here remain preliminary, they sound relevant for the field. Thus, this study appears suitable for the scope of PLOS Biology short reports. However, my enthusiasm is tempered by concerns regarding the accuracy of the gating strategy and the robustness of certain data. Please refer to my comments below for further details.

1) The authors' assertion that CD11b+Ly6G-CD45hi cells within the lesion correspond to monocyte-derived macrophages (MDMs) is conceptually flawed. The gating strategy does not exclude monocytes nor include specific staining for macrophage markers, making it impossible to definitively identify these cells as macrophages. The MDMs gate likely encompasses a mixture of MDMs, monocytes, and microglia, as activated microglia may also upregulate CD45 expression. Additionally, the authors' claim of validating their gating strategy using Ccr2-RFP reporter mice lacks support in the LPC model, as RFP expression in CD45hi cells is only demonstrated in a different model involving LPS injection in the striatum. Furthermore, the absence of analysis in sham mice or the contralateral corpus callosum makes it challenging to assess the extent of MDM infiltration within the lesion.

2) The poor accuracy of the gating strategy utilized in this study undermines the interpretation of the RNA-seq analysis presented in Fig. 2 and Fig. S3. The authors assert that RFP is still expressed in monocyte-derived macrophages (MDMs) within the lesion site. Therefore, a more accurate sorting strategy would involve gating on genuine macrophages (by carefully excluding both Ly6C+ and Ly6C- monocytes) and sorting RFP+ and RFP- cells separately. This method would offer greater accuracy than solely relying on CD45 staining. Alternatively, the authors could have utilized Ms4a3-Cre x R26-tdTomato mice, which recombine approximately ~95% of monocytes with negligible recombination in microglia.

3) The authors initially suggest that Ccr2-/- mice exhibit delayed remyelination at day 10, implying a role for monocyte-derived macrophages (MDMs) in promoting myelination. However, they later show that MDMs are enriched for WNT signature genes and that inhibiting the WNT pathway in MDMs accelerates remyelination at day 10. I find challenging to reconcile these two findings. The authors seem to propose a biphasic response of MDMs, beneficial at the onset of remyelination but potentially detrimental at later stages. While this interpretation is intriguing, it does not appear fully supported by the data presented in this manuscript. At the present stage, it is unclear whether MDMs even persist in the lesion beyond day 10.

4) Lastly, it is concerning that many critical experiments described here show statistics on n=3 mice/group, indicating that the experiments were performed only once. This raises questions about the reliability and reproducibility of the results. Increasing the sample size or replicating the experiments would improve the robustness of these findings.

Minor comments:

1) Why do the authors sometimes use the one-tailed T test instead of two-tailed? How can the authors assess the normal distribution when the N size is 3?

2) Authors stated that circulating monocytes are normally absent in the Ccr2-/- mice (page 8). Monocytes are not "absent" in these mice. They are just reduced.

3) In Fig. S3F, the authors gated DCs as CD45+CD115+Ly6Ghi cells. These are neutrophils. DCs do not express Ly6G.

4) Panel Fig. 2I displays fluorescence images of RFP+ macrophages within the lesion, but these images do not appear convincing. I can barely distinguish the shape of the cells. The low RFP signal is probably due the antigen retrieval step. I am wondering if employing Ccr2-GFP or Ms4a3-Cre : tdtomato mice could enhance the quality of these images.

Reviewer #4: 

The current submission aims to suggest that monocyte-derived macrophages and not resident microglia regulating the kinetics of remyelination in the brain. The authors inject lysophosphatidyl choline (LPC), a chemical that effectively dissolves membranes, mainly affecting myelin sheaths. This work is placed in the context of multiple sclerosis, though this model certainly lacks many aspects of that disease. The authors also present data from human samples. 

I appreciate this is a short report, but there are many issues with the data presented and are outlined below:

Figure.1 There is very little characterisation of the macrophage response to LPC.

Firstly, Fig1B shows almost no macrophages (MDM) recruited to the lesion, and at least one mouse has zero cells recruited. The representative FACS plot shows what one would expect from a naïve mouse, as there are always a few CD11b+ CD45hi cells in these preps. In an area that has been denuded of its myelin, you would expect MDM recruitment, but it certainly is not convincing here. 

The authors must show more evidence that MDMs are actually recruited to the site and that in the ccr2 -/- mice, the cells are no longer recruited. This is an absolutely crucial bit of data to verify that the downstream effects on myelin are due to absence of cells in the knockout. There are many ways to distinguish microglia and MDMs now, and the ccr2 +/rfp is one of them (as they show in Fig.2 G).

Without these data, the idea that recruited monocytes at the lesion are responsible for remyelination kinetics is difficult to support.

If it can be robustly shown that monocytes are recruited, it still does not automatically follow this MDMs affect remyelination. In E to H, how do we not know that remyelination is not just due to lack of appropriate clearance responses in the acute phase after LPC injection? 

i.e. MDMs will be recruited to the lesion as early as two days after demyelination in WT mice. In ccr2 -/- they should not be. Therefore, these results are just as likely to be due to inappropriate clearance of debris in the acute phase.

With the ccr2 -/- strategy, the authors cannot decouple inadequate clearance from regenerative response after clearance. Can the authors show that in the days 1-10 that loss of monocyte recruitment does not affect demyelination? 

For the authors to say the 'tempo' of remyelination is changed would need them to use the same measurement across time. At the moment, MAG staining is at 10dpi as an 'early remyelination marker', but how well does this correlate with myelinated axons? If they authors want to make a firm statement about remyelination 'tempo', the authors need to do a time course with between day 10 and 21 with the same readout. 

Mechanism - MDMs have a WNT signature during remyelination

A large proportion of the paper depends in bulk RNA sequencing from ultra-low RNA from sorted MDMs and microglia. Much more information is needed in order to assess the validity of these experiments as they are very prone to bias in RNA amplification steps from ultra-low material.

Firstly, what were the cell yields for the sorted MDM and sorted microglia? 

Were mice pooled? If the data in Fig. 1C are representative, more than half the mice will have less then 50 cells (some have zero). Just a few stray nuclei of neurons/astrocytes/endothelial cells and this will massively change the gene expression profiles in the sample.

Were numbers of sorted microglia equal to MDMs? If many microglia were sorted, this would reduce the noise in this group, whereas MDM group would be very much influenced by non-MDM cells.

There appears to be no biological, or even technical replicates. Was this done? Is it just one pooled sample? There is no report of n numbers. What are the paired t-tests done on? Using each gene as an n number? This clearly isn't correct if so.

The idea that Wnt is involved depends on the quality of the above data. Yet, the quantitative expression of Wnt is extremely low, in over half of the transcripts the authors suggest are significantly 'up regulated' are in fact, in their own words, negligibly expressed (FPKM<0.05) Fig S3D. 

Then, surprisingly, they appear to compare this negligible expression to that of microglia, which is zero, and therefore it is misleading to show a major fold upregulation, from microglia. 

Due to the extremely, extremely low expression of wnt, and particularly if there is only one sample, contamination from neurons and astrocytes which also express wnts is a major concern. 

The authors attempt to validate some the sequencing by showing Axin2 protein expression in lesion MDMs using Ccr2RFP/+ reporter mice and found that '85% of the RFP+ MDMs were positive (Fig.2H-I).' What time point is this at? And crucially, is axin2 protein found in microglia, or indeed other cells?

The authors now turn to human data and published sequencing data sets to examine Wnt related genes in the cells in the rim of the lesion. Some more data on the two 'classical monocyte clusters (renamed 1 and 2)' would help the reader assess the data.

The authors say these data 'revealed that ~50% of genes are known to either be regulated by WNTs or regulate WNT signalling (Fig.3A,B).' What does this mean? 50% of all genes that were measured? 50% of genes that are in the WNT pathway are regulated? What do they mean by regulated? They are expressed as fold change, but fold change from what? Are these significantly changed.

The histogram show the majority of these genes are close to Log2FC of 1, and non above 1.5. Convention suggests that Log2FC of 2 or 0.5 is necessary (with appropriate statistics including FDR etc.) to suggest a biological meaning. There is potentially non here. 

The authors also look at human blood monocytes from MS patients and five WNT-associated genes were measured, to show increases from healthy controls (Fig.3C) but how can a Paired t-test between groups done on these data? It should be a two-way ANOVA with post hoc tests.

The final figure probes the functional consequence inhibiting WNT activity in macrophages. Interestingly, they find that the floxed controls also change the 'tempo' of remyelination. 

As it 'occurred at a slower pace than expected, with poor remyelination at 14 dpl

likely resulting from the FVB background'. 

The subsequent experiments then assess the tempo of remyelination by inhibiting WNT signalling in MDMs. Some of these experiments are quite elegant, but there is pretty much no data on their characterization, again making them difficult to assess. How many treated cells were injected to each mouse? How consistent was their recruitment to the CNS?

As this is critical to the claims, can the authors also show the same effect by injecting Porc fl/fl macrophages?

Lastly, its extremely difficult to assess the tempo of something when only one time point is assessed, especially as the initially experiments show that in one phase it is delayed then later accelerated.

---

## [Decision Letter · Decision Letter 2]

3 Feb 2025

Dear Veronica,

Thank you for your patience while we considered your revised manuscript "Monocytes reduce the efficiency of central nervous system remyelination" for consideration as a Short Report at PLOS Biology. Your revised study has now been evaluated by the PLOS Biology editors, the Academic Editor and the original reviewers. 

The reviews are appended below, an as you will see, Reviewers 1, 2, and 4 are largely satisfied by the changes made in the revision (although Reviewer 4 suggests some details from the rebuttal be added to the manuscript, and we agree with that). Reviewer 3 has more substantial lingering concerns - but after discussing these with the Academic Editor, we think those points can largely be addressed with textual changes and perhaps a bit of analyses of existing data. 

Reviewer 3 has raised the concern that the study may underestimate the extent of monocyte differentiation - and we think that this is a reasonable concern. However, we would not require additional genetic fate mapping approaches to address this issue and think that it may be addressed with textual changes, and perhaps acknowledging this limitation, or toning down claims of monocyte identity. Reviewer 3 also notes a lack of enrichment of monocyte signature genes in the sorted monocyte RNA-seq dataset - and we wonder if this point could be addressed by simply presenting the FPKM data for a panel of established monocyte markers, similar to how you have presented microglia markers, in Fig 2a. 

In light of the reviews, which you will find at the end of this email, we are pleased to offer you the opportunity to address the remaining points from the reviewers in a revision that we anticipate should not take you very long. We will then assess your revised manuscript and your response to the reviewers' comments with our Academic Editor aiming to avoid further rounds of peer-review, although we might need to consult with the reviewers, depending on the nature of the revisions.

**IMPORTANT: As you address the reviewer comments, we also ask that you address the following editorial requests: 

1) TITLE: After some discussion within the team, we would like to propose an edit to the title to add a few more details from the study. If you agree (and if supported), we suggest you change the title to: 

"Monocyte-secreted Wnt reduces the efficiency of central nervous system remyelination"

We are happy for you to refine this further. 

2) ETHICS STATEMENT: Please update the ethics statement in your methods section to include the full name(s) of the IACUC/ethics committee(s) that reviewed and approved the animal care and use protocol/permit/project license. Please also include an approval number.

3) DATA: Thank you for providing your RNA-seq data on GEO. Can you provide me with a reviewer token so that I can check that this meets our requirements? Also, please note that you will need to make that data publicly available before publication (although its fine to leave it private for now). 

4) DATA: 

a. Supplementary files (e.g., excel). Please ensure that all data files are uploaded as 'Supporting Information' and are invariably referred to (in the manuscript, figure legends, and the Description field when uploading your files) using the following format verbatim: S1 Data, S2 Data, etc. Multiple panels of a single or even several figures can be included as multiple sheets in one excel file that is saved using exactly the following convention: S1_Data.xlsx (using an underscore).

b Deposition in a publicly available repository. Please also provide the accession code or a reviewer link so that we may view your data before publication. 

>>While your deposition to GEO addresses this requirement for some figures, for all other data please ensure that you provide the individual numerical values that underlie the summary data displayed in the following figure panels as they are essential for readers to assess your analysis and to reproduce it:

FIg 1C,F,H-I,K,M-R; Fig 2I-J; Fig 4C,E,I-J; Fig S1E,G-H; Fig S2C-E,G,I; Fig S3A,D,E-F;

>>Please also ensure that figure legends in your manuscript include information on where the underlying data can be found, and ensure your supplemental data file/s has a legend.

>>Please ensure that your Data Statement in the submission system accurately describes where your data can be found.

5) DATA: Please deposit the raw flow cytometry data to flow repository, or another publicly available repository. (I think Flow Repository has been giving people trouble lately, so if you encounter difficulties uploading your data, you can try Zenodo, or another repository instead).

6) CODE: Per journal policy, if you have generated any custom code during the course of this investigation, please make it available without restrictions. Please ensure that the code is sufficiently well documented and reusable, and that your Data Statement in the Editorial Manager submission system accurately describes where your code can be found.

**IMPORTANT - SUBMITTING YOUR REVISION**

*Resubmission Checklist*

*Published Peer Review*

*PLOS Data Policy*

*Blot and Gel Data Policy*

Sincerely,

Luke

Lucas Smith, Ph.D.

Senior Editor

PLOS Biology

lsmith@plos.org

REVIEWS:

Reviewer #1: The authors have fully addressed this reviewer's concerns.

Reviewer #2: All my concerns have been fully addressed. I don't have further comments. 

Reviewer #3: The authors provided a detailed point-by-point rebuttal, but this reviewer is still not satisfied. Although I do not doubt that Wnt signaling from monocytes could play some role in regulating remyelination dynamics, the manuscript contains a large number of imprecisions, and several statements are either inaccurate, misleading or wrong.

These are just a few examples of contradictory sentences:

"These findings suggest that when classical monocytes are reduced in remyelinating lesions, oligodendrocyte differentiation is impaired yet oligodendrocytes compensate by producing more myelin, suggesting that monocytes limit myelin production."

"Altogether, these findings reveal that although classical monocytes are required for the timely onset of oligodendrocyte differentiation and myelin protein expression during remyelination, they limit myelin production, thereby reducing remyelination efficiency."

I understand what the authors are trying to say, but the message is not clear at all.

"CNS lesion myeloid cells encompass resident populations (e.g. microglia) and infiltrating monocytes, which are considered to be indistinguishable from one another based on morphology and immunostaining for marker expression (12). Consequently, microglia and monocytes are often analysed as one population, with the respective contribution of each subset to remyelination being poorly understood."

This is not true. Perhaps the authors wanted to say that MDMs can become very similar to microglia upon infiltrating the CNS parenchyma.

"…our findings are in line with some of these genes having previously been shown to be expressed in monocytes albeit at a relatively lower level than in microglia (e.g. P2ry12, Tmem119), whereas Sall1 is never expressed by monocytes."

Again, not true. Monocytes don't express P2ry12 and Tmem119. This is not what the literature shows. It looks like the authors do not distinguish between monocytes and MDM (monocyte-derived macrophages).

"Surprisingly, assessment of macrophage markers by flow cytometry of wildtype mice (F4/80 expression in CD11b+CD45hiLy6C+ cells) (SFig.1F-G), or immunofluorescence analysis of Ccr2RFP/+ mice (IBA1 expression in RFP+ cells) (SFig.1H-I) indicated that approximately 90% of the cells remain undifferentiated as monocytes (F4/80-, IBA1-) in remyelinating lesions."

The authors provide zero evidence of monocytes not differentiating in the lesion. Lack of F4/80 staining in CCR2+Ly6C+ monocytes is meaningless. Monocytes start upregulating macrophage markers along with downregulation of monocyte genes. Indeed, Fig. S3C suggests that monocytes are differentiating into macrophages. If the authors want to trace the fate of monocytes within the lesion, they should use a lineage tracing tool like Ms4a3-Cre or Ccr2-CreErt2 as suggested in the initial review. The authors claimed the following: "we note that this line has non-specific activity in other immune populations." I would like to know what the authors are referring to. Ms4a3-Cre recombines in BM GMPs, thus labeling monocytes (as well as MDMs) and neutrophils. These populations can be easily distinguished by flow cytometry, so I disagree with the concern about the specificity. Perhaps the authors chose not to perform this experiment. In this case, any statements pertaining to the lack of monocyte differentiation should be removed from the manuscript as it is not substantiated by experimental evidence.

Additionally, Fig.2C shows the top 50 DEGs between microglia and monocytes. This heatmap does not contain monocytes signature genes like Ly6c2, Ccr2, Ear2, Plac8, S100a4, Vim, Lyz2, Ms4a4c, just to mention a few. Only Cd44 is consistent with the monocyte signature. At the same time, this list of DEGs contain several neuronal genes (Map2, Tubb3, etc). Based on the data here provided, there is no reason to believe that monocyte differentiation is impaired in the demyelinating lesion; without even mentioning the tons of literature demonstrating the opposite.

Reviewer #4: The data presented in the new manuscript are vastly improved. I now think that the basis of the paper is much more solid, although much of the data are not definitive and there are many open questions, the readers now have enough information to assess it for themselves. 

I would still require that information regarding the cell numbers for sequencing be included, as is described in the rebuttal:

'the average cell yield for sorted microglia was 5777 per mouse, and the average cell yield of sorted monocytes was 38 per mouse…..we sequenced an average of 4 ng of RNA for microglia and 1.6 ng of RNA from monocytes.' This is important to know for those wanting to assess the quality but also repeat the experiment should they wish.

---

## [Editor Report · Decision Letter 3]

18 Feb 2025

Dear Veronica,

Thank you for the submission of your revised Short Report "Monocyte-secreted Wnt reduces the efficiency of central nervous system remyelination" for publication in PLOS Biology and thank you for addressing the last reviewer and editorial requests in this revision. On behalf of my colleagues and the Academic Editor, Ben Emery, I am pleased to say that we can in principle accept your manuscript for publication, provided you address any remaining formatting and reporting issues. These will be detailed in an email you should receive within 2-3 business days from our colleagues in the journal operations team; no action is required from you until then. Please note that we will not be able to formally accept your manuscript and schedule it for publication until you have completed any requested changes.

PRESS

Sincerely, 

Luke

Lucas Smith, Ph.D.

Senior Editor

PLOS Biology

lsmith@plos.org